# Polyelectrolyte interactions enable rapid association and dissociation in high-affinity disordered protein complexes

Andrea Sottini [1], Alessandro Borgia[1,5], Madeleine B. Borgia[1,5], Katrine Bugge [2], Daniel Nettels[1],
Aritra Chowdhury[1], Pétur O. Heidarsson [1,6], Franziska Zosel [1,7], Robert B. Best [3✉],
Birthe B. Kragelund [2✉] & Benjamin Schuler [1,4✉]

Highly charged intrinsically disordered proteins can form complexes with very high affinity in which both binding partners fully retain their disorder and dynamics, exemplified by the positively charged linker histone H1.0 and its chaperone, the negatively charged prothymosin α. Their interaction exhibits another surprising feature: The association/dissociation kinetics switch from slow two-state-like exchange at low protein concentrations to fast exchange at higher, physiologically relevant concentrations. Here we show that this change in mechanism can be explained by the formation of transient ternary complexes favored at high protein concentrations that accelerate the exchange between bound and unbound populations by orders of magnitude. Molecular simulations show how the extreme disorder in such polyelectrolyte complexes facilitates (i) diffusion-limited binding, (ii) transient ternary complex formation, and (iii) fast exchange of monomers by competitive substitution, which together enable rapid kinetics. Biological polyelectrolytes thus have the potential to keep regulatory networks highly responsive even for interactions with extremely high affinities.

[1] Department of Biochemistry, University of Zurich, Zurich, Switzerland. [2] Structural Biology and NMR Laboratory (SBiNLab) and REPIN, Department of Biology, Ole Maaloes Vej 5, University of Copenhagen, 2200 Copenhagen, Denmark. [3] Laboratory of Chemical Physics, National Institute of Diabetes and Digestive and Kidney Diseases, National Institutes of Health, Bethesda, MD 20892-0520, USA. [4] Department of Physics, University of Zurich, Zurich, Switzerland. [5] Present address: Department of Structural Biology, St. Jude Children's Research Hospital, Memphis, TN 38105, USA. [6] Present address: Department of Biochemistry, Science Institute, University of Iceland, Dunhagi 3, 107 Reykjavík, Iceland. [7] Present address: Novo Nordisk A/S, Novo Nordisk Park, 2760 Måløv, Denmark. ✉email: robert.best2@nih.gov; bbk@bio.ku.dk; schuler@bioc.uzh.ch

Interactions between proteins are at the core of cellular regulation. A remarkably large fraction of proteins involved in transcription, molecular self-assembly, and signaling in higher eukaryotes contain disordered regions that are involved in recognition and binding processes[1]. Although in many cases their detailed roles are still elusive, increasing evidence indicates that these intrinsically disordered proteins (IDPs) considerably extend the repertoire of biomolecular interaction mechanisms[2,3]. Examples include the formation of flexible protein scaffolds that can act as hubs and integrate signals by binding to many interaction partners simultaneously[4]; the ability of IDPs to assume different folded structures upon binding to different targets, which increases the number of specific interactions even for small proteins[5]; the high accessibility for post-translational modifications[6]; and the presence of multivalent interactions[7]. Structural disorder can mediate even very large assemblies of biomolecules, with properties very different from those of simple binary interactions, such as the permeability barrier of the nuclear pore complex[8] or mesoscopic cellular condensates[9].

At the structural level, biomolecular complexes involving IDPs cover a broad and virtually continuous spectrum, from cases where binding is coupled to folding, resulting in highly structured complexes, to cases where segments or entire proteins remain disordered in the complex[10,11]. We recently identified an extreme case of such a disordered protein complex: the two highly and oppositely charged human proteins prothymosin α (ProTα, net charge −44) and linker histone H1.0 (H1, net charge +53) interact with picomolar affinity (at 165 mM ionic strength) but retain their disorder in the bound state[12,13]. Experimental data from single-molecule, circular dichroism, and nuclear magnetic resonance (NMR) spectroscopy of the complex can be explained by molecular simulations that describe the two proteins as polyelectrolyte chains with the charge distribution of the respective protein sequences but without specific binding sites. The result is a highly dynamic ensemble of rapidly interconverting configurations dominated by electrostatic interactions (Fig. 1a). Here we show that these properties of the complex have important consequences for the interaction mechanism and kinetics: they enable not only rapid, diffusion-limited association, but also fast, concentration-dependent dissociation, despite the very high affinity. The underlying process of competitive substitution[14] via short-lived ternary complexes may be a means of keeping regulatory networks in the cell highly responsive even if individual binding partners have extreme affinities—a mechanism that could be widespread among IDP interactions.

## Results

**Binding equilibria of a high-affinity polyelectrolyte complex.**
To be able to probe the binding between ProTα and H1 from picomolar to micromolar concentrations, we used confocal single-molecule Förster resonance energy transfer (FRET) spectroscopy of freely diffusing molecules. By attaching a donor and an acceptor fluorophore to one of the two proteins, the binding of the unlabeled partner can be monitored by an increase in transfer efficiency, since the mutual charge screening of the oppositely charged IDPs leads to chain compaction[12]. The dissociation constant ($K_D$) of the ProTα-H1 (PH) complex increases from a few picomolar to ~1 nM when the ionic strength of the solution is increased from 165 to 200 mM (Supplementary Fig. 1)[12], within the range commonly considered physiological[15]. To reduce the complications caused by adhesion of H1 to surfaces[12], we performed all measurements at 200 mM ionic strength unless stated otherwise. Titration of 50 pM ProTα labeled at positions 56 and 110 with Alexa Fluor 488 and 594 as donor and acceptor, respectively, shows a clear transition from the unbound to the

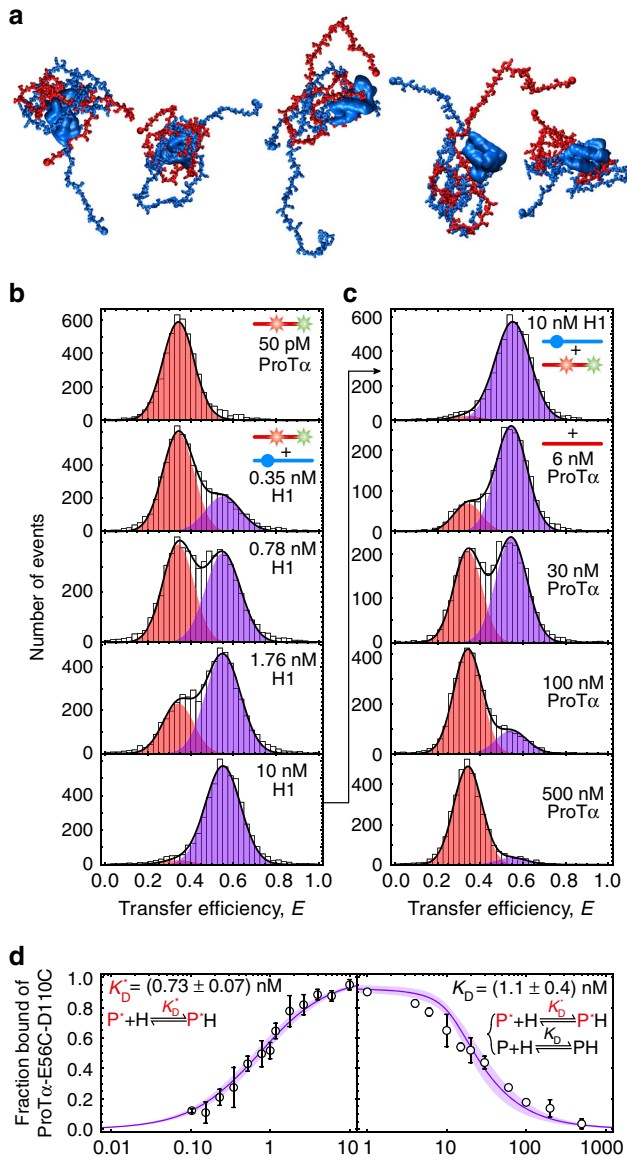

**Fig. 1 High-affinity binding of H1 to ProTα is independent of fluorescence labeling. a** Illustration of the highly dynamic and intrinsically disordered protein complex between ProTα (red) and H1 (blue) with snapshots from coarse-grained MD simulations[12]. **b** Transfer efficiency histograms of 50 pM ProTα E56C/D110C labeled with Alexa Fluor 488/594 in the presence of increasing concentrations of unlabeled H1 at 200 mM ionic strength, fit globally with two Gaussian peak functions for the unbound (red) and bound (purple) ProTα populations, respectively (sum: black lines). **c** Transfer efficiency histograms from a competition experiment with constant concentrations of 50 pM labeled ProTα and 10 nM unlabeled H1, and increasing concentration of unlabeled ProTα as indicated in the panels. **d** Resulting fractions of bound labeled ProTα as a function of the total concentration of H1 (left panel, **b**) and unlabeled ProTα (right panel, **c**). The global fit of the two datasets (continuous line, see "Methods" for details) results in an affinity of H1 for fluorophore-labeled ProTα of (0.73 ± 0.07) nM and for unlabeled ProTα of (1.1 ± 0.4) nM (s.d. calculated from at least three independent repeats).

bound population with increasing concentrations of unlabeled H1 (Fig. 1b), yielding a $K_D$ of 0.73 ± 0.07 nM (Fig. 1d).

To assess the effect of dye labeling of ProTα on the affinity, we performed a competition titration (Fig. 1c): starting from 50 pM

labeled ProTα saturated with 10 nM unlabeled H1, the labeled ProTα was outcompeted by increasing amounts of unlabeled ProTα (up to 0.5 μM). If labeled and unlabeled ProTα had identical affinities to H1, the midpoint for the competition would be expected at 18.5 nM unlabeled ProTα (see "Methods" for details on the solution of the coupled binding equilibria). The observed midpoint of ~20 nM (Fig. 1d) yields a $K_D$ of 1.1 ± 0.4 nM for the complex between the unlabeled binding partners. Similarly, measurements with ProTα and H1 labeled with different fluorophores[12], and at different positions in the sequence show that such individual amino acid exchanges and fluorescence labeling alter the $K_D$ by not more than about a factor of three (Supplementary Table 1 and Fig. 1). The perturbation of the affinity by the fluorophores is thus small, at most on the order of thermal energy ($k_BT$), in contrast to recent speculative concerns[16]. For comparison, a similar effect on binding free energy results from a change in solution ionic strength by a mere 10–20 mM.

The titrations in Fig. 1 indicate two-state-like behavior, with slow exchange between bound and unbound ProTα compared to the averaging time of the measurement, in this case given by the average fluorescence burst duration of about a millisecond, which is governed by the diffusion time through the confocal detection volume. Assuming a simple two-state binding equilibrium between ProTα (P) and H1 (H), P + H ⇌ PH, with a $K_D$ of ~1 nM and the diffusion-limited association rate coefficient, $k_{on}$, of ~$10^9$ $M^{-1}$ $s^{-1}$,[12] we expect a dissociation rate coefficient, $k_{off} = k_{on}K_D$, of ~1 $s^{-1}$. The exchange rate between unbound and bound ProTα is $k_{ex} = k_{on}c_H + k_{off}$, where $c_H$ is the concentration of unbound H1. Owing to the low concentrations of unbound H1 present at the conditions used in Fig. 1, the reaction is thus indeed in the regime of slow exchange between bound and unbound ProTα (see Supplementary Fig. 2 for the detailed concentration dependences of $k_{ex}$). At the midpoint of the titration in Fig. 1c, for instance, we expect slow exchange with $k_{ex} \approx 1$ $s^{-1}$ using the values above.

This behavior changes, however, if the titrations are performed at higher protein concentrations, as shown in Fig. 2 in direct comparison with the competition experiment from Fig. 1c. In the presence of 1 μM unlabeled H1, the titration with unlabeled ProTα exhibits a single transfer efficiency peak that shifts continuously, with some broadening at intermediate ProTα concentrations, indicating intermediate to fast exchange (Fig. 2b). At 20 μM H1, a single peak of uniform intensity at all ProTα concentrations indicates fast exchange between bound and unbound populations on the millisecond timescale (Fig. 2c). Similarly, fast exchange with only moderate line broadening is observed in $^1$H–$^{15}$N heteronuclear single-quantum-coherence (HSQC) spectra of 20 μM $^{15}$N-labeled ProTα titrated with unlabeled H1 (Fig. 2d–f). In NMR, the relevant timescale for the detection of separate signals for the bound and unbound populations is determined by the resonance frequency difference between them, in our case up to ~20 Hz, i.e., ~50 ms. If we assume a two-state binding process, this fast exchange observed both in the single-molecule and the NMR experiments is in apparent contradiction[17] to the low $k_{off}$ and $K_D$ inferred from the single-molecule experiments in the low nanomolar protein concentration range (Fig. 1). This point is illustrated by NMR lineshape calculations for the reaction P + H ⇌ PH using the rate coefficients from the fluorescence experiments at low concentrations, which result in two well-separated peaks at fixed chemical shifts (Fig. 2g). The nanomolar concentration regime is not accessible by NMR, but the similarity of the observations in the single-molecule and NMR experiments at micromolar protein concentrations indicates that the changes in exchange behavior with concentration is intrinsic to the molecular system rather

than being method-dependent. The goal of this work is to elucidate the underlying molecular mechanism.

**Binding kinetics of a high-affinity polyelectrolyte complex.** From the free-diffusion measurements presented so far, only information about the exchange rate, i.e., the sum of the association and dissociation rates of the observed species, can be inferred. To separate the two contributions and probe the kinetics of the ProTα–H1 complex over a range of protein concentrations, we used single-molecule FRET of immobilized molecules. Labeled and biotinylated ProTα was immobilized on a polyethylene glycol-passivated surface and monitored by confocal fluorescence detection in the presence of nanomolar concentrations of unlabeled H1 (Fig. 3a–c). As expected for a simple 1:1 binding reaction, the resulting fluorescence time traces exhibit transitions between two states, bound (high-FRET efficiency) and unbound (low-FRET efficiency). The average dwell time of ProTα in the unbound state decreases with increasing H1 concentration, corresponding to the expected increase in association rate. To quantify $k_{on}$ and $k_{off}$, we analyzed the time traces by likelihood maximization using a two-state hidden Markov model (see "Methods"). From the slope of the observed association rate as a function of H1 concentration, we obtained $k_{on} = (1.45 ± 0.06) × 10^9$ $M^{-1}$ $s^{-1}$ (Fig. 3c), confirming the diffusion-limited association observed in stopped-flow measurements[12], and in accord with a downhill-binding reaction that does not require activated structural rearrangements or configurational search. The average dwell time in the bound state, corresponding to the inverse dissociation rate, shows the hallmark of a simple two-state binding reaction: it is independent of H1 concentration (Fig. 3c). The resulting $K_D$ of 1.2 ± 0.1 nM is very similar to the value measured for freely diffusing molecules (Fig. 1), indicating that surface immobilization does not interfere with binding.

As in the free-diffusion experiments (Fig. 2), we next probed the influence of higher protein concentrations on the kinetics (Fig. 3d–f). At a constant concentration of 10 nM unlabeled H1, adding up to 100 nM unlabeled ProTα leads to an obvious decrease in the dwell times of both the bound and the unbound states. Figure 3f shows the resulting association and dissociation rates as a function of concentration. Qualitatively, a decrease in the on-rate with increasing ProTα concentration is expected because ProTα scavenges H1 in solution and reduces its availability for binding to the labeled immobilized ProTα. However, the observed drop is much less pronounced than expected for a simple two-state reaction involving only the unbound proteins (H, P) and their 1:1 complex (PH) (dashed line in Fig. 3f). The deviation from this simple two-state model is even more obvious for the off-rate, which increases about 30-fold between 0 and 100 nM unlabeled ProTα rather than remaining constant. These results demonstrate that a two-state binding model is insufficient for describing the interaction between H1 and ProTα at concentrations above the low nanomolar range.

**Ternary complex formation accelerates exchange.** The protein concentration-dependent dissociation rates we observed are a characteristic signature of binding processes that involve the formation of transient ternary complexes[18]. Ternary complex formation is often facilitated for IDPs owing to the pronounced dynamics in their complexes and the possibility of multivalent interactions[4,19–21]. Higher-order complex formation is expected to be particularly likely for interactions between highly disordered polyelectrolytes, such as ProTα and H1[12–14,22,23]. Indeed, several lines of evidence indicate that their association does not terminate at the 1:1 complex if an excess of either binding partner is added: (i) donor/acceptor-labeled H1 bound to ProTα exhibits a decrease

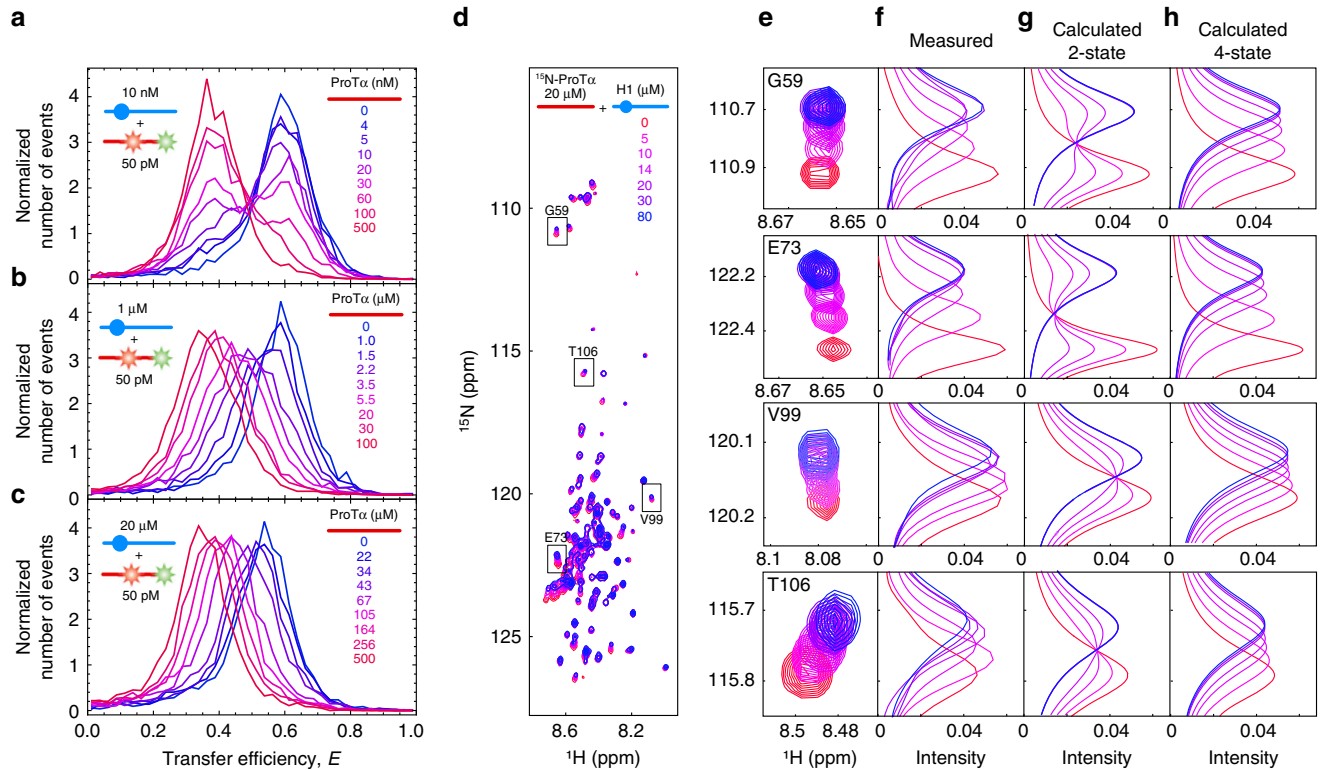

**Fig. 2 Association/dissociation kinetics are highly protein concentration-dependent. a–c** Overlay of transfer efficiency histograms of 50 pM ProTα E56C/D110C labeled with Alexa Fluor 488/594 at different concentrations of unlabeled H1 (10 nM, 1 μM, and 20 μM) titrated with increasing concentration of unlabeled ProTα exhibit a transition from slow to fast exchange between the populations (see legends for color code). The mean transfer efficiency of bound ProTα decreases slightly with increasing H1 concentration owing to the formation of higher-order complexes (Supplementary Fig. 3d). **d** ¹H-¹⁵N-HSQC spectra of 20 μM ¹⁵N-ProTα (recorded at 750 MHz and 200 mM ionic strength) titrated with increasing concentrations of unlabeled H1 (see legend for color code). **e** Examples of individual resonances illustrate the continuous shifting of a single population-averaged peak. **f** 1D ¹⁵N projections of the NMR resonances at different concentrations of unlabeled H1. **g, h** Comparison of NMR 1D ¹⁵N lineshapes calculated using the Bloch–McConnell equation for a two-state (**g**) or a four-state binding mechanism (**h**) (see Supplementary Table 2 for rate coefficients and "Methods" for details).

in transfer efficiency if an excess of unlabeled ProTα in the micromolar range is added[12] (Supplementary Figs. 3 and 4). (ii) Concurrently, the hydrodynamic radius of the complex increases, as measured by fluorescence correlation spectroscopy (FCS) and NMR (Supplementary Fig. 3). (iii) Finally, coarse-grained molecular simulations also reveal the formation of stable higher-order complexes[12].

Given the excess of ProTα in the measurements in Figs. 1 and 2, the most likely ternary complex is formed by the association of an additional ProTα molecule (P) with an existing ProTα–H1 complex (PH), resulting in ProTα₂–H1 (PPH). Including the ternary complex as a third state in the kinetic model can indeed explain the pronounced protein concentration dependencies of the observed association and dissociation rates (Fig. 3f). However, accurately quantifying the underlying rate coefficients requires an extended range of ProTα concentrations that is difficult to probe with single-molecule surface experiments alone. We therefore combined different types of measurements and analysis to extend the kinetics to micromolar ProTα concentrations (Fig. 4). A way of accessing faster timescales in single-molecule FRET experiments is the use of recurrence analysis of single particles (RASP)[24], which allows the kinetics of interconversion between subpopulations to be determined from equilibrium free-diffusion measurements (see "Methods"). In the present case, relaxation times can be quantified down to ~1 ms, corresponding to exchange rates of up to ~1000 s⁻¹ (Fig. 4a and Supplementary Fig. 5). To test the consistency with ensemble experiments, we further performed stopped-flow measurements where the

preformed complex of donor/acceptor-labeled ProTα and unlabeled H1 is mixed with unlabeled ProTα, and the relaxation to the equilibrium distribution of bound vs. unbound labeled ProTα is observed (Fig. 4b). Figure 4c summarizes the combined kinetic results and illustrates the consistency of the different methods, over a wide range of protein concentrations, and covering exchange rates from ~10 to ~1000 s⁻¹. The exchange rates deviate from the behavior expected for a simple two-state binding reaction by up to three orders of magnitude at high ProTα concentrations (Fig. 4c), thus clearly revealing a more complex interaction mechanism.

To analyze the combined kinetic data quantitatively in terms of a model including the ternary complex PPH, we used the rate coefficients for the reaction H + P ⇌ PH obtained at very low protein concentrations, where ternary complex formation is negligible (Fig. 3a, b). This leaves us with two free fit parameters: $k_{on}^{PPH}$, the association rate coefficient of ternary complex formation (PH + P → PPH), and $k_{off}^{PPH}$, its dissociation rate coefficient (PPH → PH + P). (Note that we neglect the formation of the ternary complex PHH because of the large excess of ProTα present; we also neglect the dissociation pathway PPH → PP + H because the ProTα dimer is highly disfavored by charge repulsion.) From the fit of the combined kinetic data (Fig. 4c), we obtained $k_{on}^{PPH} = (0.53 \pm 0.04)\cdot10^9$ M⁻¹ s⁻¹—in the diffusion-limited range, and only slightly lower than the value for the binary complex. With $k_{off}^{PPH} = (1.9 \pm 0.2)\cdot10^3$ s⁻¹, however, dissociation is almost three orders of magnitude faster for the PPH

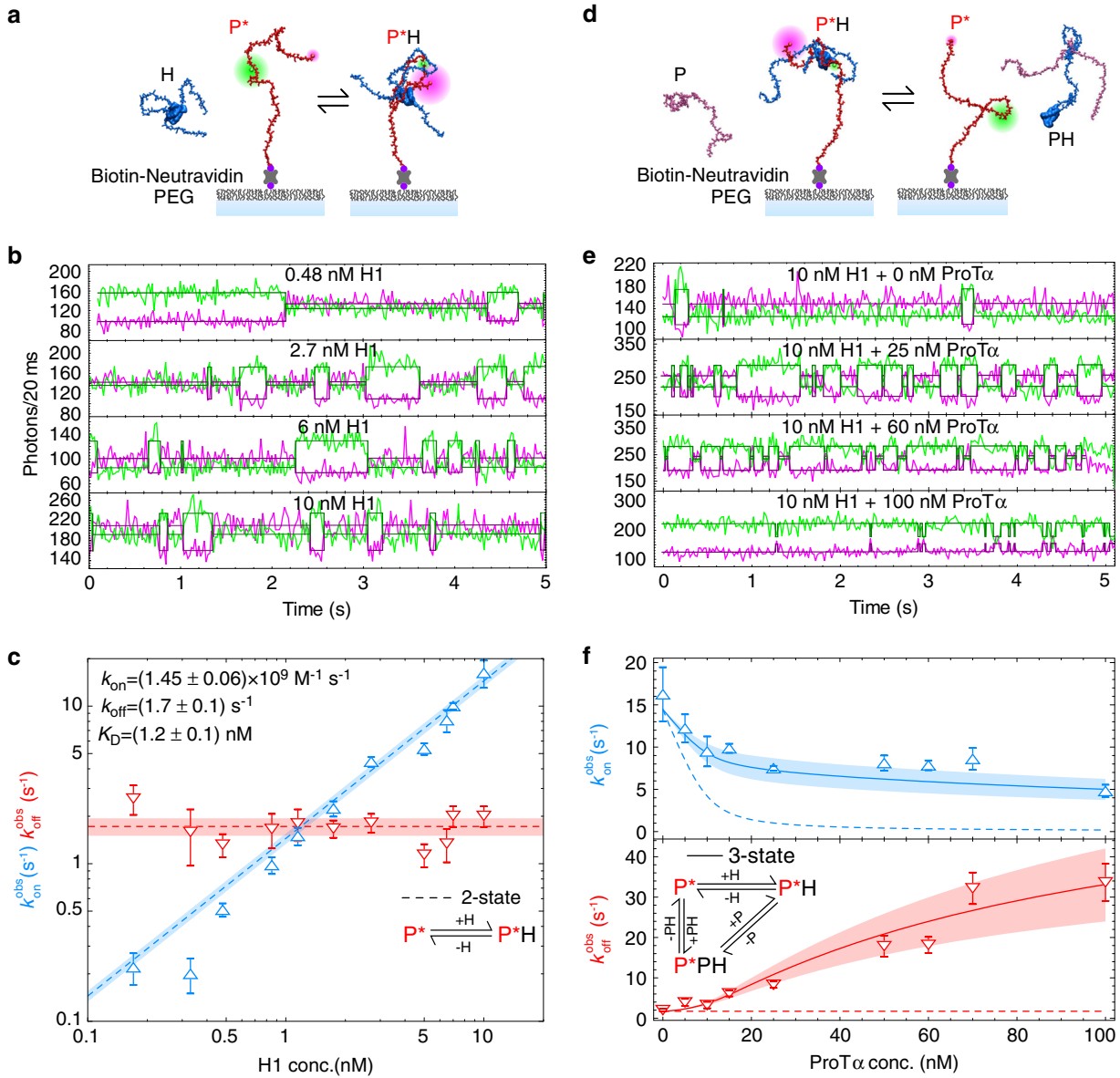

**Fig. 3 Ternary complex formation explains binding kinetics. a** Cartoon representation of surface-immobilized Avi-Tag-ProTα E56C/D110C labeled with Cy3B/LD650 (P*, red) binding to H1 (H, blue). **b** Representative single-molecule fluorescence time traces in the presence of increasing concentrations of unlabeled H1 in solution (concentrations indicated in panels). The most likely state trajectories (bound/unbound) identified using the Viterbi algorithm are depicted as magenta and green lines. **c** Observed association ($k_{on}^{obs}$, blue upward triangles) and dissociation ($k_{off}^{obs}$, red downward triangles) rates as a function of the concentration of unlabeled H1. The fit based on a two-state binding model (dashed lines) results in the rate coefficients $k_{on}$ = (1.45 ± 0.06)·$10^9$ $M^{-1} s^{-1}$ and $k_{off}$ = (1.7 ± 0.1) $s^{-1}$. **d** Cartoon representation of competition between surface-immobilized labeled ProTα bound to H1 with unlabeled ProTα (P, pink). **e** Experiment analogous to (**b**), but in the presence of 10 nM unlabeled H1 and different concentrations of unlabeled ProTα, as indicated in the panels. **f** $k_{on}^{obs}$ and $k_{off}^{obs}$ as a function of ProTα concentration with a fit (solid lines) to a three-state model including the ternary complex PPH (see inset) and the dependencies expected for the two-state model shown in (**c**) (dashed lines) (see "Methods" for details). s.d. in (**c**) and (**f**) are from bootstrapping (see "Methods" for details).

trimer than for the PH dimer. The ternary complex thus has a lifetime of only ~0.5 ms, which adds an efficient kinetic channel for the rapid interconversion between bound and unbound ProTα via PPH (Fig. 3f inset). Since the formation of this short-lived ternary complex is favored by high protein concentrations, it explains the strong increase in the observed dissociation rate with increasing protein concentration that is responsible for the accelerated exchange rate compared to two-state binding (Fig. 4c, d). Remarkably, a population of only ~2% PPH accelerates dissociation 30-fold under the conditions of Fig. 3f. For the protein concentrations in the single-molecule FRET experiments

in Fig. 2c, the resulting exchange rates range between ~$10^3 s^{-1}$ and $10^4 s^{-1}$ (Supplementary Fig. 2), explaining the observation of a single transfer efficiency peak shifting continuously with increasing ProTα concentration.

If the presence of a short-lived ternary complex can explain the kinetics observed by fluorescence, it should also allow us to rationalize the fast-exchange behavior observed by NMR. In both their bound and unbound states, ProTα and H1 rapidly sample myriad disordered configurations or relative arrangements on the 20- to 100-ns timescale[12,25]. The chemical shifts in either state thus represent averages of the sampled microscopic local

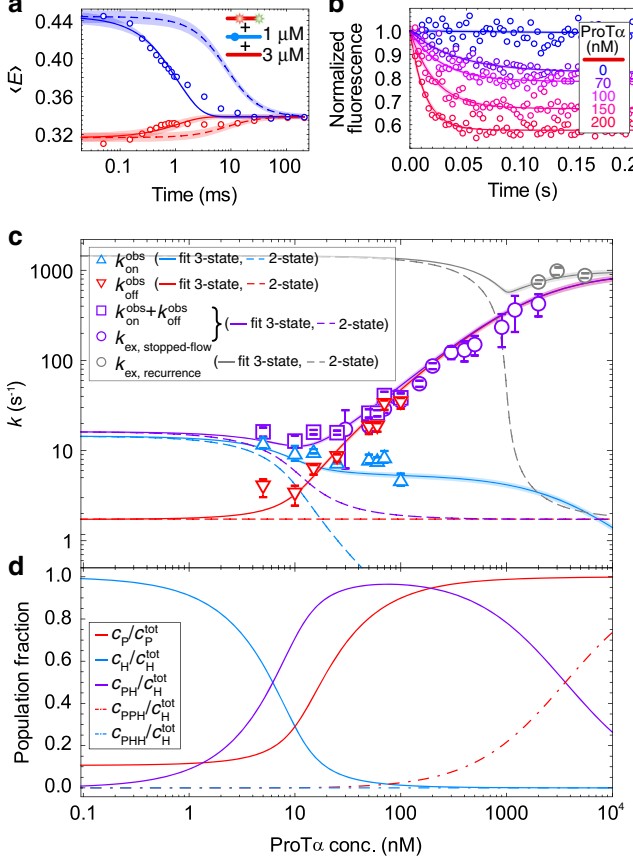

**Fig. 4 Ternary complex formation explains kinetics over a broad range of concentrations. a** Kinetic recurrence analysis[24] of 50 pM ProTα E56C/D110C Alexa Fluor 488/594 with 1 μM unlabeled H1 and 3 μM unlabeled ProTα, showing the bound (blue) and unbound (red) states at different delay times with global fit (solid lines) using a single relaxation rate, resulting in $k_{ex} = (864 \pm 77)$ s$^{-1}$. Dashed lines show the contribution to the relaxation due to the occurrence of new molecules in the confocal volume. **b** Normalized acceptor fluorescence signal from stopped-flow ensemble kinetics upon rapid mixing of 2 nM ProTα E56C/D110C Alexa Fluor 488/594 and 10 nM unlabeled H1 in a 1:10 ratio with buffer (blue) or solutions containing increasing concentrations of unlabeled ProTα (purple to red, see color scale in legend). The fluorescence signals are fit with single-exponential decay functions to quantify the relaxation rate corresponding to $k_{ex}$. **c** Global fit of kinetic data from different methods as a function of the total concentration of ProTα (in the presence of 10 nM H1). Observed association rates ($k_{on}^{obs}$) and dissociation rates ($k_{off}^{obs}$) from FRET experiments with surface-immobilized molecules can be summed to obtain the exchange rate (purple squares) for direct comparison with $k_{ex}$ from stopped-flow experiments (purple circles) and from kinetic recurrence analysis (gray circles). Solid lines show the global fit of all data using a model including the formation of PPH (see "Methods", Eq. (14)); note that recurrence analysis was performed at increased H1 concentration, resulting in higher $k_{ex}$); dashed lines show the dependencies of the observed association (blue), dissociation (red), and exchange rates (purple, gray) expected for a two-state binding model. s.d. of single-molecule data (red and blue triangles, purple squares) are from bootstrapping (see "Methods"), s.d. of stopped-flow data (purple circles) are from at least fifty repeats each, with standard errors of the fits shown. s.d. of recurrence data (gray circles) are from three independent replicates. **d** The fraction of each population (relative to the total ProTα concentration, $c_P^{tot}$, for P; relative to the total H1 concentration, $c_H^{tot}$ (10 nM), for H, PH, PPH, PHH) calculated using the rate coefficients from the global fit as a function of the concentration of unlabeled ProTα (see "Methods", Eq. (13)).

environments. The resulting average chemical shifts, however, are different in the bound and the unbound states, and the interconversion between them is governed by the association/dissociation kinetics. To describe the NMR data at the requisite micromolar $^1$H-$^{15}$N-ProTα concentrations (Fig. 2d), we further need to consider that the titration ranges from a large excess of ProTα to a large excess of H1. In view of the symmetry expected for the ternary complex formation of two highly charged IDPs[12], we thus also have to consider two H1 molecules bound to one ProTα molecule (PHH, Supplementary Fig. 4). The $K_D$ we obtain for PH + H ⇌ PHH (12 ± 3 μM) is slightly higher than for PH + P ⇌ PPH ($K_D = 3.5 \pm 0.4$ μM), as expected from the positive net charge of the ProTα-H1 complex[12], yet in a similar range.

Using the values of the rate coefficients obtained from the combined kinetic experiments (Fig. 4 and Supplementary Table 2), we can reconstruct the NMR titrations. Figure 2e, f shows examples of measured NMR resonances with sufficiently large chemical shift perturbations and lack of overlap with other signals extracted for a detailed analysis. We solved the Bloch–McConnell equations[26] for the corresponding kinetic model, including PPH and PHH (see "Methods") to calculate the expected changes in NMR lineshape along the $^{15}$N axis (where the shifts are most pronounced) as a function of protein concentration. The close correspondence of the calculated spectra (Fig. 2h) with the experimental results supports the proposed kinetic mechanism and reconciles the slow-exchange behavior in single-molecule results at pico- to nanomolar protein concentrations with the fast-exchange behavior seen in the NMR measurements at micromolar protein concentrations. Note that at the typical concentrations of H1 and ProTα used in an NMR measurement of the 1:1 complex[12], the ternary complexes are difficult to detect directly, since they comprise only a small fraction of the total population, and their chemical shifts are close to those of the dimer (Supplementary Fig. 6). For instance, at 20 μM of each H1 and ProTα (the lowest feasible concentration range for high-quality NMR spectra), the total concentration of ternary complexes amounts to only ~0.7 ± 0.1 μM, but they result in an acceleration of the exchange rate by a factor of ~46 compared to two-state binding (Supplementary Fig. 2).

**Competitive substitution enables rapid exchange.** To obtain insights into the molecular mechanism underlying the rapid interconversion between bound and unbound ProTα, we turned to molecular dynamics simulations using a coarse-grained model that has previously been shown to provide good agreement with the conformational, dynamic, and kinetic properties of the ProTα–H1 complex[12]. Simulations of ProTα–H1 in the presence of an additional ProTα molecule illustrate how the pronounced disorder in the binary complex facilitates ternary complex formation (Fig. 5a). From the ternary complex, either one of the two ProTα molecules is then expected to dissociate with equal probability because conformational equilibration in the complex occurs on the 100-ns timescale[12], much faster than the millisecond lifetime of the complex, resulting in a separation of timescales and Markovian behavior[27]. For half of these ternary complex formation and dissociation events, the net result is therefore the displacement of one ProTα molecule in the complex by another (see Supplementary Movies 1 and 2 for examples illustrating each case), a process previously referred to as "competitive substitution" in the context of synthetic polyelectrolytes[14]. In view of the large net charge of PPH, it is not surprising that, according to our kinetic analysis, the average lifetime of the ternary complex is three orders of magnitude shorter than that of the PH dimer (Fig. 4). Consequently, events

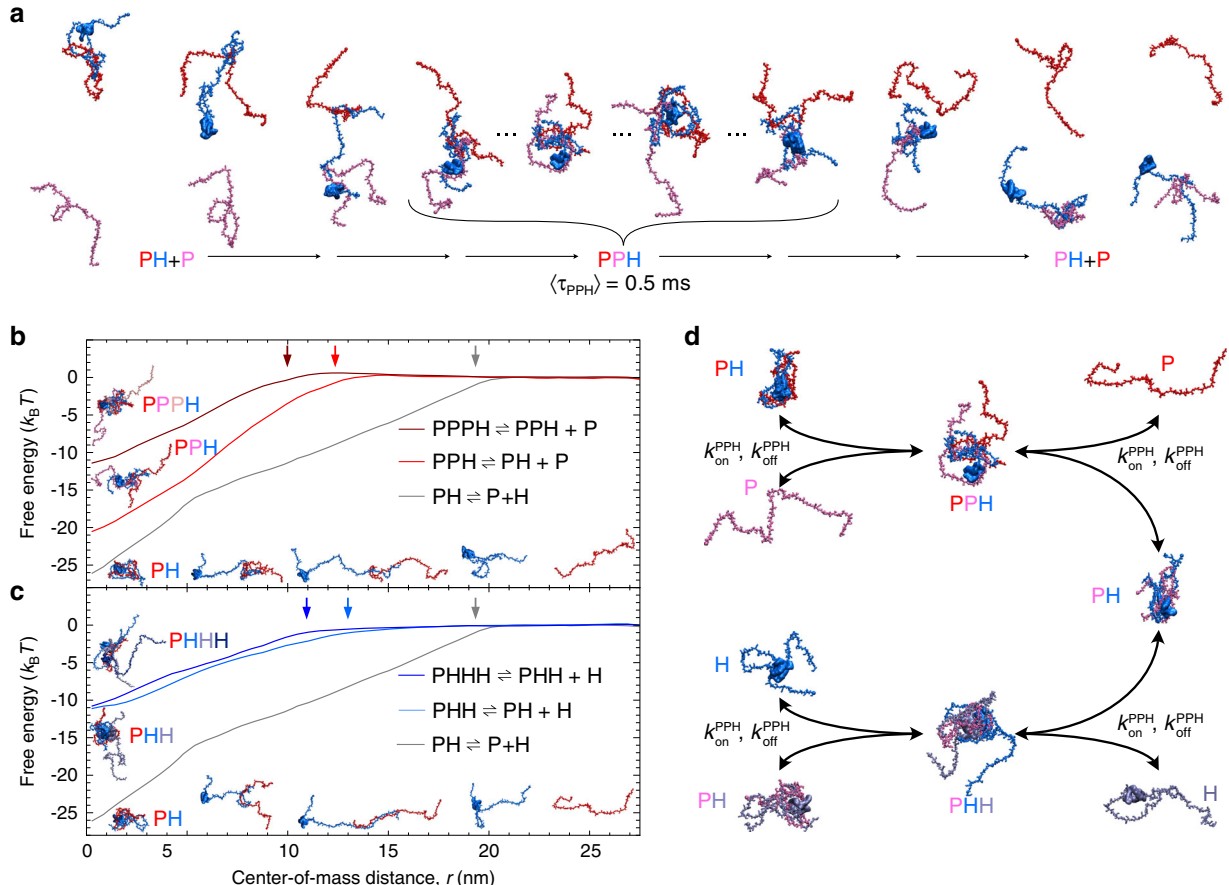

**Fig. 5 Polyelectrolyte interaction mechanisms from molecular simulations. a** Snapshots from coarse-grained molecular dynamics simulations (see Supplementary Movie 1) of a competitive substitution event via formation of a transient ternary complex PPH (average PPH lifetime, $\langle\tau_{PPH}\rangle \approx 0.5$ ms). A ProTα molecule (P, pink) associates with a preformed ProTα-H1 complex (PH), shares contacts with H1 (H, blue), and eventually replaces the ProTα molecule previously bound (P, red). The alternative scenario of association and dissociation of the same molecule of ProTα is shown in Supplementary Movie 2. **b, c** Potentials of mean force for the formation of the PH dimer, PPH and PHH trimers, and PPPH and PHHH tetramers (see legend) from umbrella sampling simulations, with snapshots of the oligomers (left) and of the formation of PH from P and H at different distances of their centers of mass, $r$ (bottom). Approximate capture radii for the different complexes are indicated with colored arrows. **d** Kinetic scheme of dimer and trimer formation, including competitive substitution (color scheme as in **a**).

such as the one shown in Fig. 5a lead to an acceleration of the exchange of ProTα molecules between bound and unbound states, observed experimentally as an increase in exchange rate with increasing protein concentrations, where ternary complex formation is more and more favored (Figs. 3 and 4).

The potentials of mean force obtained from the simulations provide further mechanistic insight into the binding mechanism and the two essential contributions to the molecular basis of the fast kinetics of polyelectrolyte interactions (Fig. 5b, c). First, they show that the association between ProTα and H1 is essentially a downhill process, as reflected by the absence of an activation barrier and, correspondingly, a diffusion-limited association rate coefficient estimated from the translational diffusion coefficients of the monomers and the simulated potential (Supplementary Fig. 7). The simulations further suggest that ProTα and H1 already interact when the distance between their centers of mass is ~20 nm (Fig. 5b, c), much greater than the sum of their average hydrodynamic radii (~6 nm)[12], owing to their rapid sampling of very extended conformations, representing an extreme example of fly-casting[28]. This large capture radius leads to a more than 20-fold increase in the rate of reaching their average center-of-mass distance of ~1 nm relative to the rate of an unbiased diffusive encounter (Supplementary Fig. 7).

It is worth pointing out that even in the absence of electrostatic attraction, binding remains a downhill process and thus in the diffusion-limited regime (Supplementary Fig. 7), in accord with the small effect of salt concentration on the experimentally observed association rate (Supplementary Fig. 8). This behavior contrasts with folded proteins, where electrostatic steering in the encounter complex can lead to very large rate enhancements by reducing the fraction of non-productive binding events[29,30]. The absence of an activation barrier for the association of disordered polyelectrolytes can be explained by the lack of conformational changes or orientational search required for binding: initial contact formation is possible between virtually any segment of the negatively charged ProTα and any segment of the positively charged H1. Once in contact, the two chains essentially slide into each other and maximize their electrostatic interaction (Fig. 5b, c), which is reached in a large ensemble of rapidly interconverting and mutually adapting configurations[12] (Fig. 1a). The pronounced dependence of the $K_D$ on salt concentration[12] (Supplementary Fig. 1) is thus almost entirely due to changes in the dissociation rate coefficient (Supplementary Fig. 8).

The second important aspect that is reflected by the potentials of mean force (Fig. 5b, c) is the reduced stability of the ternary complexes compared to the 1:1 complex, which results in the

highly increased dissociation rates of the ternary complexes and underlies the accelerated exchange between bound and unbound molecules via competitive substitution. Notably, however, the association rate coefficient for trimer formation estimated from the potentials is only approximately twofold lower than for dimer formation, close to the experimentally observed ratio of 2.7 ± 0.5. Binding of an additional molecule to the 1:1 complex thus remains a downhill process, albeit less electrostatically favored and with a smaller capture radius since the chains are more compact in the dimer (Fig. 5b, c).

The simulations further support the symmetry of the process of ternary complex formation between H1 and ProTα, i.e., that PPH is formed with an excess of ProTα, and PHH with an excess of H1 (Fig. 5b, c and Supplementary Fig. 7). As expected from the charge imbalance between H1 (net charge +53) and ProTα (net charge −44), the affinity of the ProTα–H1 dimer for another ProTα molecule is predicted to be greater than for H1, in agreement with experiment (Supplementary Table 1). Competitive substitution can thus occur both via PPH or PHH, depending on the relative concentrations of the two binding partners. Moreover, the simulations indicate the existence of higher oligomers beyond the ternary complex. Although these species have so far eluded detailed experimental investigation, the continuous change of transfer efficiencies and hydrodynamic radii with an excess of one binding partner extending into the millimolar range (Supplementary Fig. 3) are suggestive of this behavior. In view of the similarity of the transfer efficiencies of PH and PHH extrapolated to 200 mM ionic strength (Supplementary Fig. 4), it is in fact very likely that changes in transfer efficiency at the high excess concentrations of one binding partner (Supplementary Fig. 3) are caused by larger oligomers.

## Discussion

For folded biomolecules, on-rate coefficients for complex formation are typically below ~$10^6 \, M^{-1} \, s^{-1}$, and high affinities are attained by slow dissociation[29]. In contrast to this behavior, our results show how the extremely high-affinity interaction between charged IDPs can lead to rapid kinetics that exhibit a pronounced dependence on protein concentration. Both association and dissociation kinetics are strongly affected by the disorder in the individual binding partners and the complex. Association between the highly positively charged H1 and the highly negatively charged ProTα is downhill in free energy, and the large capture radius combined with the electrostatic interaction between the highly expanded and dynamic chains enables on-rate coefficients above $10^9 \, M^{-1} \, s^{-1}$. At very low protein concentrations, simple two-state kinetics are observed, and dissociation rates between ~$10^{-3} \, s^{-1}$ and $1 \, s^{-1}$ reflect the picomolar to nanomolar affinities in the physiological ionic-strength range (~165–200 mM). At higher protein concentrations, however, an additional H1 or ProTα molecule can associate with a ProTα–H1 dimer (Fig. 5). Since the transient ternary complexes formed in this process have $K_D$ values orders of magnitude higher than the 1:1 complex and only millisecond lifetimes, one of the H1 or ProTα molecules will dissociate rapidly. The result is a much more rapid exchange between bound and unbound molecules than at low protein concentrations. The corresponding kinetic model (Fig. 5d) describes the behavior of the system over a broad range of conditions and explains the underlying change in kinetic mechanism from the picomolar protein concentrations accessible only with single-molecule experiments to the high micromolar protein concentrations probed by both fluorescence and NMR.

We emphasize that our results confirm that the very high affinity between H1 and ProTα is independent of the presence of the fluorophores used for single-molecule FRET[12] (Fig. 2). A very high affinity between H1 and ProTα had previously been reported semiquantitatively[31]; however, the picomolar to nanomolar affinities in the 1:1 complex combined with the extreme tendency of H1 to adhere to the surfaces of reaction tubes and sample chambers[12] has so far prevented quantitative affinity measurements in this concentration range with methods other than single-molecule spectroscopy. Measurements with techniques such as calorimetry[16], which require much higher protein concentrations, will inherently include a contribution from the ternary complexes formed in the presence of an excess of one binding partner; they can further be affected by liquid–liquid phase separation[13,22]; and they are complicated by the low or even positive reaction enthalpy and the important role of counterion-release entropy as the driving force of polyelectrolyte interactions[12,32,33].

Although the formation of transient ternary complexes has a strong effect on the kinetics of polyelectrolyte interactions, the 1:1 ProTα–H1 complex is the dominant species at sub-micromolar protein concentrations and even at equimolar protein concentrations in the high micromolar range[12] (Supplementary Fig. 3c, f, h). Even though the ternary complexes are orders of magnitude less stable than the 1:1 complex, the association of highly charged IDPs is unlikely to terminate strictly at the stage of the ternary complex, and both simulations (Fig. 5b, c) and experiments (Supplementary Fig. 3) indicate that larger oligomers may form. Their existence would affect the concentration dependence of the observed rates at very high protein concentrations quantitatively, but the qualitative mechanism of competitive substitution is expected to be operational for any size of oligomers[14]. Under certain conditions, such as high protein concentrations and low ionic strength, it is conceivable that a continuum of oligomers may form, ultimately leading to liquid–liquid phase separation[9] in the form of coacervates, which are commonly observed for synthetic polyelectrolytes and polypeptides[23,34,35] and have recently been reported for H1 and ProTα[13]. Competitive substitution within such mesoscopic assemblies may contribute to their stability and functional intracellular roles.

The kinetic mechanism described here is conceptually related to the facilitated dissociation or "molecular stripping" that has been reported for a range of multi-domain DNA- or RNA-binding proteins[36–41] and antibody–antigen binding[42], which can have important regulatory consequences and also involve multivalent interactions that enable ternary complex formation[18]. The electrostatically driven interactions of biological polyelectrolyte complexes, with their lack of specific binding sites or motifs[12,13,22,43,44], could be considered an extreme case of multivalent interactions that are particularly conducive to ternary complex formation and facilitated dissociation via competitive substitution[14]. Biological polyelectrolyte complexes are thus well suited for mediating high-affinity interactions and yet enabling efficient biological regulation via rapid molecular handoffs between different biomolecular assemblies. Important potential examples involving charged IDPs and nucleic acids are abundant in the nucleus and, among other processes, essential for chromatin remodeling[21,22]. It is worth noting that both H1 and ProTα are present at micromolar concentrations in the nucleus[45,46], which suggests that the role of ternary complex formation is relevant for their cellular interactions. Related interaction mechanisms connected to deviations from simple two-state kinetics have also been reported for less charged IDPs[19]. Moreover, the association of IDPs is often faster than for folded proteins since it involves lower activation barriers[47,48], and multivalent interactions are common for IDPs[4,19–21], suggesting that kinetic mechanisms similar to those of H1 and ProTα may be more widespread and important for cellular regulation than previously thought.

## Methods

**Protein preparation**. The variant of H1 for fluorescent labeling was produced by bacterial expression using a modified version of the pRSET vector[49] containing the cysteine variant of the human *H1F0* gene (*UniProt P07305*), a hexahistidine tag, and a thrombin cleavage site[12]. The H1 variant was expressed in *E. coli* C41 cells[50] using a terrific broth medium at 37 °C, induced with 0.5 mM isopropyl-β-D-1-thiogalactopyranoside (IPTG) at an OD600 of ~0.6, and grown for 3 further hours. Cell pellets were collected and resuspended in denaturing buffer (6 M guanidinium chloride, GdmCl) in phosphate-buffered saline (PBS, 10 mM sodium phosphate pH 7.4, 137 mM NaCl, 2.7 mM KCl), the soluble fraction was collected and applied to a Ni$^{2+}$-IDA resin (ABT Beads) in batch. The resin was washed twice with five resin volumes of denaturing buffer including 25 mM imidazole, three times with five resin volumes of PBS including 25 mM imidazole, and the protein was eluted with PBS including 250 or 500 mM imidazole. The protein was dialyzed against PBS, filtered and its hexahistidine tag cleaved off with 5U of thrombin (Serva) per milligram of H1, for 2 h at room temperature. Uncleaved protein and the tag were removed on an immobilized metal affinity chromatography (IMAC) column (HisTrap HP 5-ml GE Healthcare) in PBS, including 25 mM imidazole. H1 was further purified using a Mono S ion-exchange chromatography column (GE Healthcare), washed with 20 mM Tris (pH 8.0), including 200 mM NaCl, and eluted in 20 mM Tris (pH 8.0) buffer with a gradient from 200 mM to 1 M NaCl. Finally, samples for labeling were reduced with 20 mM dithiothreitol and purified by reversed-phase high-performance liquid chromatography (RP-HPLC) on a Reprosil Gold C4 column (Dr. Maisch) with an elution gradient from 5% acetonitrile and 0.1% trifluoroacetic acid in aqueous solution to 100% acetonitrile. H1-containing fractions were lyophilized and resuspended in degassed 6 M GdmCl, 50 mM sodium phosphate buffer (pH 7.0). Unlabeled recombinant wild-type human histone H1.0 was from New England Biolabs (cat. # M2501S).

For experiments using biotinylated ProTα, the cDNA of the human *PTMA* gene (*UniProt P06454*) was cloned into a pAT222-pD expression vector[51] (using forward primer: GGG CGG ATC CGG CAG CAT GTC AGA CGC AGC CG and reverse primer: GAT GAG AAG CTT GGC TAC GGC TGC CAC GCG GAC CGC CGC AAT CCT CGT CGG TC), yielding an expression construct with an N-terminal Avi-tag and a thrombin-cleavable C-terminal His$_6$-tag. pBirAcm (avidity) was co-transfected for in vivo biotinylation of Lys12 in the Avi-tag. All ProTα variants (see Supplementary Table 1 and Fig. 1 for a complete list of variants) were expressed in *E. coli* C41 cells and terrific broth medium at 37 °C. In total, 50 μM biotin in 10 mM bicine buffer (pH 8.3) (for the biotinylated variant) and 1 mM IPTG were added to the culture at an OD600 of ~0.6 and grown at 37 °C for three more hours. The cells were harvested by centrifugation and lysed by sonication. His$_6$-tagged proteins in the soluble fraction were enriched via an IMAC Ni$^{2+}$-Sepharose column (HisTrap Excel, GE Healthcare). The His$_6$-tag was then cleaved off with HRV 3 C protease or Thrombin, depending on the variant, and removed by another round of IMAC. Finally, the protein was separated from impurities by RP-HPLC on a Reprosil Gold 200 C18 column (Dr. Maisch) with a gradient from 5% acetonitrile and 0.1% trifluoroacetic acid in aqueous solution to 100% acetonitrile, lyophilized, and stored at −80 °C. All protein sequences are shown in Supplementary Table 3.

For NMR spectroscopy, ProTα was uniformly labeled with $^{15}$N by growing cells in M9 minimal medium containing $^{15}$NH$_4$Cl as the sole source of nitrogen and then purified essentially as described above[12]. The concentration of ProTα was determined by BCA assay (Thermo Fisher Scientific). For producing H1 for NMR spectroscopy, DNA-encoding human histone H1.0 was cloned into a pET-24b expression vector yielding an expression construct with a cleavable N-terminal His$_6$-tagged SUMO-tag. The SUMO-H1 plasmid was transformed into *E. coli* BL21 (DE3) cells and grown in batches of 1 l LB medium to an OD600 of ~0.6–0.8, followed by induction with 0.3 mM IPTG. Cells were harvested by centrifugation after 3 h of expression at 37 °C and lysed by sonication in 6 M GdmCl, 25 mM imidazole, PBS pH 8 (25 ml). His$_6$-tagged proteins in the soluble fraction were enriched using a 5 ml gravity-flow Ni$^{2+}$-Sepharose column (GE Healthcare), with 1 h incubation followed by washing with 6 M GdmCl, 25 mM imidazole, PBS buffer (pH 8) (50 ml), and elution by 6 M GdmCl, 500 mM imidazole, PBS pH 8 (40 ml). To the eluate we added 10 μg ml$^{-1}$ DNase and dialyzed against 2.5 mM MgCl$_2$ and 0.5 mM CaCl$_2$, 10 mM sodium phosphate buffer (pH 7). The dialysate was loaded onto a HiTrap Heparin HP column (GE Healthcare), and eluted with a linear gradient of 0–100% 2 M NaCl over 40 column volumes (CVs). The His$_6$-tagged SUMO-tag was subsequently cleaved by adding 0.2 μg ULP1 (purified as described by Singh and Graether[52]) per 1 μg of SUMO-H1, and the solution dialyzed against 200 mM NaCl, 2 mM DTT, 50 mM Tris-HCl (pH 8). Finally, the H1 sample was brought to 0.1% TFA and loaded onto a Zorbax C18 (Agilent) RP-HPLC column, eluted by a stepwise gradient of 0.08% TFA, 70% acetonitrile (2 CV 0–45%, 2 CV 45–60%, 2 CV 60–100%). H1 fractions were lyophilized and stored at −20 °C. Proper refolding of the globular domain of H1 was confirmed via $^{15}$N-HSQC spectra.

**Fluorophore labeling**. For double-labeling H1, both dyes (dissolved in dimethylsulfoxide) were added to the protein in a 1:1:1 molar ratio. Reactions were incubated at room temperature for 2 h, and stopped by adding 20 mM dithiothreitol.

Products were purified by RP-HPLC[12]. Lyophilized Avi-tagged ProTα E56C/D110C was dissolved under nitrogen atmosphere at a concentration of 200 μM in 100 mM potassium phosphate buffer, pH 7.0. The protein was then labeled for 3 h at room temperature with a 0.7:1 molar ratio of Cy3B maleimide (GE Healthcare) to protein. Labeled protein was separated from unlabeled protein by RP-HPLC on a Reprosil Gold C18 column (Dr. Maisch). The second labeling reaction was carried out for 3 h at room temperature with a 2:1 molar ratio of LD650 maleimide[53] (Lumidyne) to protein. For preparing doubly labeled variants of ProTα, the protein in 100 mM potassium phosphate buffer, pH 7.0, was incubated with Alexa Fluor 488 maleimide (Invitrogen) at a dye-to-protein ratio of 0.8:1 for 1 h at room temperature and then with Alexa Fluor 594 maleimide (Invitrogen) at 1:1 molar ratio overnight at 4 °C. The labeled protein was separated and purified by RP-HPLC on a Reprosil Gold C18 column (Dr. Maisch). The correct mass of the labeled protein was confirmed by electrospray ionization mass spectrometry.

**Single-molecule fluorescence spectroscopy**. Single-molecule fluorescence experiments were performed with a MicroTime 200 confocal microscope (Pico-Quant, Berlin, Germany) equipped with a 488-nm diode laser (Sapphire 488–100 CDRH, Coherent, Santa Clara, CA) and an Olympus UplanApo 60x/1.20 W objective. After passing through a 100-μm pinhole, sample fluorescence was separated into donor and acceptor components using a dichroic mirror (585DCXR, Chroma, Rockingham, VT). After passing appropriate filters (Chroma ET525/50 M, HQ650/100), each component was focused onto avalanche photodiodes (SPCM-AQR-15, PerkinElmer Optoelectronics, Vaudreuil, QC, Canada), and the arrival time of every detected photon was recorded (HydraHarp 400, PicoQuant, Berlin, Germany). The 488-nm diode laser was set to an average power of 100 μW at the sample. The laser was operated in continuous-wave mode for recurrence experiments (Supplementary Fig. 5) and for H1 titration experiments at lower ionic strength (Supplementary Fig. 4), or in pulsed mode for pulsed interleaved excitation of the dyes. Other labeling variants of ProTα were measured on a MicroTime 200 equipped with a 532-nm continuous-wave laser (LaserBoxx LBX-532-50-COL-PP, Oxxius) adjusted to provide ~100 μW for free-diffusion experiments, or 1–2 μW for surface experiments, at the sample. Fluorescence was separated from scattered light with a triple-band mirror (zt405/530/630rpc, Chroma), and a long-pass filter (532 LP Edge Basic, Chroma) was used to separate the 532-nm laser light from the emitted fluorescence. The fluorescence was then focused on a 100-μm pinhole and split onto two channels with a dichroic mirror (T635LPXR, Chroma). Donor emission was filtered with an ET585/65 m bandpass filter (Chroma), acceptor emission with a RazorEdge LP 647 RU long-pass filter (Chroma), detected with avalanche photo-diode detectors (SPCM-AQR-15, PerkinElmer, Waltham MA, USA), and photon arrival times recorded with a HydraHarp 400 event timer (PicoQuant). For sample scanning, the objective (UplanApo 60/1.20 W; Olympus, Japan) was mounted on a combination of two piezo-scanners, a P-733.2CL for XY-positioning and a PIFOC for Z-positioning (Physik Instrumente, Germany).

In experiments on freely diffusing molecules, transfer efficiencies were quantified from all selected photon bursts (at least 3000 bursts), each originating from an individual molecule diffusing through the confocal volume, according to $E = n_A/(n_A + n_D)$, where $n_D$ and $n_A$ are the numbers of donor and acceptor photons in each burst, respectively, corrected for background, channel crosstalk, acceptor direct excitation, differences in quantum yields of the dyes, and detection efficiencies[54]. For experiments on surface-immobilized molecules, adhesive silicone hybridization chambers (Secure Seal Hybridization Chambers, SA8R-2.5, Grace Bio-Labs) to PEGylated, biotinylated glass coverslips (22 × 22 mm Premier Bio_01, MicroSurfaces, Inc.) to form 150-μl reaction chambers. 0.2 mg ml$^{-1}$ neutravidin in 50 mM phosphate buffer was incubated in the chamber for 5 min, followed by three washing steps with the same buffer. A solution of 10 pM doubly labeled Avi-tagged ProTα was added to the chamber, incubated for 5 min and unbound protein removed by three washing steps to obtain a surface coverage of 0.05-0.2 molecules per μm$^2$. All data analysis was conducted using a custom Wolfram Symbolic Transfer Protocol (WSTP) add-on for Mathematica (Wolfram).

**Single-molecule binding equilibria**. To test whether the presence of the dyes influences the binding affinity of ProTα to H1, we conducted two kinds of titration experiments on freely diffusing molecules. Both types of experiments were performed in 10 mM Tris-buffered saline (TBS) and 0.1 mM EDTA, with the ionic strength adjusted using KCl, 140 mM β-mercaptoethanol (Sigma Aldrich) for photoprotection, and 0.01% Tween 20 (Thermo Scientific) to minimize surface adhesion. In the first kind, the dissociation constant of the labeled variant of ProTα (P*), $K_D^*$, was determined as described previously[12]: a set of FRET efficiency histograms was recorded at different concentrations of unlabeled H1 (H) and globally fit with two Gaussian peak functions corresponding to the subpopulations of unbound P* and the complex P*H, respectively. The positions and widths of the peaks were shared fit parameters, while the amplitudes were fit individually for each histogram. From the relative peak areas, we obtained the fraction of P* bound to H1 (P*H) as a function of the total H1 concentration. These data were then fit

with the binding isotherm of the reaction

$$P^* + H \overset{K_D^*}{\rightleftharpoons} P^*H,\tag{1}$$

which is

$$\theta(c_H^{tot}) = \frac{\left(c_H^{tot} + K_D^* + c_{P^*}^{tot}\right) - \sqrt{\left(c_H^{tot} + K_D^* + c_{P^*}^{tot}\right)^2 - 4c_H^{tot}c_{P^*}^{tot}}}{2c_{P^*}^{tot}},\tag{2}$$

where $c_H^{tot}$ and $c_{P^*}^{tot}$ are the total concentrations of unlabeled H1 and labeled ProTα, respectively.

In the second kind of experiment, we kept the concentration of labeled ProTα and unlabeled H1 constant and varied the concentration of unlabeled ProTα (P) to compete out the labeled ProTα. From the resulting sets of measured transfer efficiency histograms, we obtain, analogous to the H1 titration, the fraction of labeled ProTα bound to H1, but now as a function of $c_P^{tot}$, the total concentration of unlabeled ProTα. For the two coupled reactions

$$P^* + H \overset{K_D^*}{\rightleftharpoons} P^*H$$
$$P + H \overset{K_D}{\rightleftharpoons} PH\tag{3}$$

we find the fraction of P* bound to H as

$$\theta_{coupled}(c_P^{tot})$$
$$= \frac{c_H^{tot}K_D - c_P^{tot}K_D^* - 2c_H^{tot}K_D - K_D^*K_D + K_D^*\sqrt{(-c_H^{tot} + K_D + c_P^{tot})^2 + 4c_H^{tot}K_D}}{2\left[c_H^{tot}(K_D^* - K_D) + K_D^*(-c_P^{tot} - K_D + K_D^*)\right]}.\tag{4}$$

This relation holds for $c_{P^*}^{tot} \ll c_P^{tot}$, which is the case even for the lowest $c_P^{tot}$ used. Under the conditions of Fig. 1d (right panel), with $c_H^{tot} = 10$ nM, the midpoint of the fraction of bound P* would be reached at $c_P^{tot} = 18.5$ nM if $K_D^* = K_D$. As $K_D^*$ is known independently from the H1 titration described above, $K_D$ is the only adjustable parameter and can be determined from fitting Eq. (4) to the competition data. We repeated the measurements for different concentrations of H (ranging from 1 to 50 nM). Throughout, the concentration of P* was 50 pM in TBS buffer at different ionic strengths (adjusted by varying the KCl concentration). The measured dissociation constants of all variants are compiled in Supplementary Table 1 and compared in Supplementary Fig. 1. The derivation of equations and fits of the experimental data were done using Mathematica (Wolfram Research).

To quantify the equilibrium dissociation constant of the ternary complex PHH, $K_D^{PHH}$, we titrated unlabeled H1 to 50 pM labeled ProTα fully saturated with unlabeled H1, in TBS at low ionic strength (from 8 mM to 83 mM), where it is possible to resolve the PHH subpopulation in transfer efficiency histograms (Supplementary Fig. 4). The system is described by the binding equilibrium

$$P^*H + H \overset{K_D^{PHH}}{\rightleftharpoons} P^*HH.\tag{5}$$

To extract $K_D^{PHH}$, we fit $\theta$, the fraction of the P*HH complex relative to the total concentration of P*, $c_{P^*}^{tot} = c_{P^*H} + c_{P^*HH}$, as a function of the total concentration of H1 with

$$\theta(c_H^{tot}) = \frac{(c_H^{tot} + K_D^{PHH} + c_{P^*}^{tot}) - \sqrt{(c_H^{tot} + K_D^{PHH} + c_{P^*}^{tot})^2 - 4c_H^{tot}c_{P^*}^{tot}}}{2c_{P^*}^{tot}}.\tag{6}$$

Note that the concentration of unbound ProTα, $c_{P^*}$, is negligible under the experimental conditions.

**Hydrodynamic radii from fluorescence correlation spectroscopy (FCS).** The fluorescence intensity crosscorrelations, $G(\tau)$, between donor and acceptor signal were fit with a model including translational diffusion and triplet blinking to determine the diffusion time, $\tau_D$, through the confocal volume:

$$G(\tau) = 1 + \frac{1}{N}\left(1 + \frac{\tau}{\tau_D}\right)^{-1}\left(1 + s^2\frac{\tau}{\tau_D}\right)^{-1/2}\left(1 + c_T\exp\left[-\frac{\tau}{\tau_T}\right]\right),\tag{7}$$

where $N$ is the average number of labeled molecules in the confocal volume, $s$ is the ratio of the lateral to the axial radii of the confocal volume, and $\tau_T$ is the correlation time of triplet blinking. The observed hydrodynamic radius, $R_H$, of the protein complex as a function of the concentration of either ProTα or H1, was obtained from

$$R_H = \tau_D\frac{R_H^{PH}}{\tau_D^{PH}},\tag{8}$$

where $\tau_D$ is the diffusion time measured with FCS, and $R_H^{PH}$ and $\tau_D^{PH}$ are the hydrodynamic radius and the diffusion time of the 1:1 complex, respectively. For $R_H^{PH}$, we used the value previously measured with 2-focus FCS[12].

**Kinetic analysis of experiments on surface-immobilized molecules.** All single-molecule experiments on surface-immobilized molecules were performed under Argon atmosphere in degassed TBS buffer using[55] D$_2$O instead of H$_2$O with different ionic strengths (adjusted with KCl). 1.5 nM protocatechuate 3,4-dioxygenase (PCD, Sigma Aldrich) and 2 mM protocatechuic acid (PCA, Sigma Aldrich) were added as an oxygen scavenging system, as well as 1 mM methyl viologen and 1 mM ascorbic acid as triplet-state quenchers. Single immobilized molecules were localized by scanning a $20 \times 20$ μm$^2$ area ($256 \times 256$ pixels). The molecular brightness was optimized by adjusting the objective collar and the $z$ position of the focus. Time traces from immobilized molecules were recorded until one of the dyes photobleached. Data were acquired with custom-developed software. Note that a kinetic analysis probing PHH formation in single-molecule surface experiments was not possible because double-labeled H1 has not been amenable to surface immobilization.

**Analysis of photon time traces.** Single-molecule photon time traces were inspected to ensure that no substantial brightness variations had occurred (e.g., by a drift of the molecule's position relative to the confocal volume, long-lived dark states, or background variation). Traces with stable signal and more than four transitions were analyzed until photobleaching of either the donor or the acceptor dye occurred. Single-step photobleaching indicated that only one molecule was present in the confocal volume. The time traces reveal transitions between two states of low and high-FRET efficiency, corresponding to unbound and bound ProTα, respectively. Hence, we describe the system with two states, whose interconversion is described by the rate matrix

$$\mathbf{K}_2 = \begin{pmatrix} -k_{on}^{obs} & k_{off}^{obs} \\ k_{on}^{obs} & -k_{off}^{obs} \end{pmatrix},\tag{9}$$

where $k_{on}^{obs} = k_{on}c_H$ is the observed pseudo-first-order association rate coefficient; $c_H$ is the concentration of unbound H1 in solution (in this experiment, given the small amount of surface-immobilized ProTα, $c_H = c_H^{tot}$ to very good approximation); $k_{on}$ is the second-order association rate coefficient. $k_{off}^{obs}$ is the observed pseudo-first-order dissociation rate coefficient. In case of the surface experiments with H in the low nanomolar range, $k_{off}^{obs} = k_{off}$, the first-order dissociation rate coefficient. The rate coefficients and the photon detection rates were determined using the maximum-likelihood approach described by Gopich and Szabo[56]. The likelihood for time trace $j$ is calculated from the general equation

$$L_j = \mathbf{1}^T\prod_{i=1}^{N_j}\mathbf{n}_{c_i,j}\exp\left[\left(\mathbf{K} - \mathbf{n}_{D,j} - \mathbf{n}_{A,j}\right)\tau_i\right]\mathbf{p}_{eq},\tag{10}$$

where $N_j$ is the total number of photons in the time trace; $c_i$ is the color of the $i$th photon (D or A); $\tau_{i=1} = 0$, and $\tau_{i>1}$ is the inter-photon time, i.e., the time interval between the detection of the $(i-1)$th and $i$th photon. $\mathbf{K} = \mathbf{K}_2$ is the rate matrix from Eq. (9). $\mathbf{n}_{D,j}$ and $\mathbf{n}_{A,j}$ are diagonal matrices with the observed donor photon rates $\left(n_{D,j}^U \quad n_{D,j}^B\right)$ and the acceptor photon rates $\left(n_{A,j}^U \quad n_{A,j}^B\right)$ of the two states on the diagonal, respectively. The photon rates vary slightly from time trace to time trace, mainly because the immobilized molecules are located at slightly different positions inside the laser focus. $\mathbf{1}^T = \begin{pmatrix} 1 & 1 & \cdots \end{pmatrix}$ is the transposed vector of ones. $\mathbf{p}_{eq}$ is the equilibrium population vector of states given by $\mathbf{K}\mathbf{p}_{eq} = 0$ and $\mathbf{1}^T\mathbf{p}_{eq} = 1$. To estimate the most likely set of parameters describing the data, we maximized $L = \sum_j \ln\left(L_j\right)$, the logarithm of the total likelihood for all photon time traces, with respect to $k_{off}^{obs}$, $k_{on}^{obs}$, $\mathbf{n}_{D,j}$, $\mathbf{n}_{A,j}$. For each single-molecule measurement of surface-immobilized molecules, at least 40 time traces were used, and the analysis was performed on a minimum of 200 individual association or dissociation events (although usually many more). Standard deviations for the results of each dataset were estimated by 20 rounds of bootstrapping.

**Recurrence analysis.** Measurements for recurrence analysis[24] were performed with 488-nm continuous-wave excitation adjusted to 300 μW at the sample to maximize photon emission. Solutions of ProTα labeled with Alexa Fluor 488/594 and 1 μM H1 in TBS (with the ionic strength adjusted to 200 mM with KCl) were mixed with 2–5.5 μM unlabeled ProTα. To obtain sufficient burst statistics, data acquisition times were in the range of 10–24 h. Photon bursts from single molecules diffusing through the confocal volume were identified in two steps. First, sequences of consecutive photons separated by less than 30 μs were combined in a single burst. Second, bursts containing more than 100 photons were selected and then split into 100-μs intervals. Only the intervals containing more than 50 photons were retained for recurrence analysis[24]. The resulting burst list (>500,000 bursts for each experiment) was scanned for burst pairs in which the first burst matches a given transfer efficiency range (initial $E$ range) and the second was detected within a given time interval after the first. From the set of second bursts, a recurrence histogram of transfer efficiencies was generated. The initial $E$ ranges (indicated in the histograms, Supplementary Fig. 5) were chosen to correspond to values either mostly populated by the bound species, PH, or the unbound species, P. The center values of the time intervals in which the second bursts fall, i.e., the recurrence times, were varied from 0.1 ms to 500 ms. The resulting recurrence

histograms were then fit with a log-normal peak function to describe the asymmetric donor-only population and a Gaussian peak function to describe the FRET population. The positions of the Gaussian peaks are plotted as a function of the recurrence time, $t$, and fit with

$$\langle E\rangle(t) = (1 - p_{\text{same}}(t))\langle E\rangle_{\text{eq}} + p_{\text{same}}(t)\left[\langle E\rangle_{\text{eq}} + \left(\langle E\rangle(0) - \langle E\rangle_{\text{eq}}\right)e^{-k_{\text{ex}}t}\right], \quad (11)$$

where $\langle E\rangle_{\text{eq}}$ is the transfer efficiency at infinite delay time, $\langle E\rangle(t=\infty)$; $\langle E\rangle(0)$ is the transfer efficiency $t=0$ ms; $k_{\text{ex}}$ is the exchange rate; and $p_{\text{same}}(t)$ is the probability that the second burst is emitted by the same molecule as the first burst, which can be calculated from the data independently[24]. For each concentration of unlabeled ProTα, two to three independent repeats were performed and used to estimate the average $k_{\text{ex}}$ and the standard deviation.

**Stopped-flow kinetics.** Rapid-mixing ensemble fluorescence experiments were carried out with an Applied Photophysics PiStar-180 stopped-flow spectrometer. A solution containing 2 nM ProTα double-labeled with Alexa Fluor 488/594 (at positions 56 and 110) and 10 nM H1 was mixed 1:10 with a solution containing unlabeled ProTα (yielding final concentrations of unlabeled ProTα of 30 nM, 50 nM, 60 nM, 70 nM, 90 nM, 100 nM, 150 nM, 200 nM, 300 nM, 400 nM, 500 nM, 900 nM, 1.2 μM, and 2 μM). The decrease in acceptor fluorescence emission resulting from the expansion of labeled ProTα upon dissociation from H1 was used to monitor the reaction by exciting at 436 nm (10 nm bandwidth) using a HgXe lamp and recording the fluorescence emission passing a 580-nm long-pass filter. The buffer used was TBS with a total ionic strength of 200 mM (adjusted with KCl) in the presence of 0.01% Tween 20 to minimize surface adhesion of the proteins. For each concentration of ProTα, the observed rate $k_{\text{ex,stopped-flow}}$ and its standard error were estimated by fitting the averaged signal from at least 50 repeats.

**Global analysis of concentration-dependent kinetics.** The observed pseudo-first-order kinetic rate coefficients were obtained from a global analysis of data from single-molecule surface experiments, stopped-flow measurements, and recurrence measurements performed with labeled ProTα in the presence of unlabeled H1 and increasing concentrations of unlabeled ProTα (Fig. 4). In addition to the formation of the PH complex, we need to include the formation of the ternary complexes, PPH and PHH:

$$P + H \underset{k_{\text{off}}}{\overset{k_{\text{on}}}{\rightleftharpoons}} PH$$
$$P + PH \underset{k_{\text{off}}^{\text{PPH}}}{\overset{k_{\text{on}}^{\text{PPH}}}{\rightleftharpoons}} PPH \quad . \quad (12)$$
$$PH + H \underset{k_{\text{off}}^{\text{PHH}}}{\overset{k_{\text{on}}^{\text{PHH}}}{\rightleftharpoons}} PHH$$

Given the similarity of the trimer association rate coefficients based on the potentials of mean force obtained from the umbrella sampling simulations (Supplementary Fig. 7), we assume $k_{\text{on}}^{\text{PHH}} = k_{\text{on}}^{\text{PPH}}$ for our analysis.

Labeled ProTα (P*) can thus populate four states, P*, P*H, P*PH, and P*HH; the formation of P*P*H is negligible because $c_{\text{P}^*}^{\text{tot}} \ll c_{\text{P}}^{\text{tot}}$. The kinetics for the observation of P* (i.e., for experiments where P is fluorescence- or isotope-labeled) are described by the rate matrix

$$\mathbf{K}_{4,\text{P}} = \begin{pmatrix} -k_{\text{on}}c_{\text{H}} - k_{\text{on}}^{\text{PPH}}c_{\text{PH}} & k_{\text{off}} & k_{\text{off}}^{\text{PPH}}/2 & 0 \\ k_{\text{on}}c_{\text{H}} & -k_{\text{off}} - k_{\text{on}}^{\text{PPH}}c_{\text{P}} - k_{\text{on}}^{\text{PHH}}c_{\text{H}} & k_{\text{off}}^{\text{PPH}}/2 & k_{\text{off}}^{\text{PHH}} \\ k_{\text{on}}^{\text{PPH}}c_{\text{PH}} & k_{\text{on}}^{\text{PPH}}c_{\text{P}} & -k_{\text{off}}^{\text{PPH}} & 0 \\ 0 & k_{\text{on}}^{\text{PHH}}c_{\text{H}} & 0 & -k_{\text{off}}^{\text{PHH}} \end{pmatrix}. \quad (13)$$

$c_{\text{P}}$, $c_{\text{H}}$, and $c_{\text{PH}}$ are determined by the three equilibrium constants (Eq. (12)) and the total concentrations of P and H, $c_{\text{P}}^{\text{tot}}$ and $c_{\text{H}}^{\text{tot}}$. (Note that the matrix is independent of $c_{\text{P}^*}$, $c_{\text{P}^*\text{H}}$, $c_{\text{P}^*\text{PH}}$, and $c_{\text{P}^*\text{HH}}$ in the limit of $c_{\text{P}^*}^{\text{tot}} \ll c_{\text{P}}^{\text{tot}}$, the model is linear in these concentrations. Further, we assume that P*PH can dissociate to P* + PH or to P*H + P with equal likelihood, leading to the factors ½ in the corresponding matrix elements.) In the single-molecule time traces (Fig. 3), we can only distinguish a low-FRET and a high-FRET state. The low-FRET state corresponds to P*; the other three states (P*H, P*PH, and P*HH) are indistinguishable at 200 mM ionic strength and exhibit high FRET (Supplementary Fig. 4e). From the measurements, we can extract the mean dwell times of the low-FRET state, $\tau_{\text{low}}^{\text{P}}(c_{\text{P}}^{\text{tot}})$, and of the high-FRET state, $\tau_{\text{high}}^{\text{P}}(c_{\text{P}}^{\text{tot}})$, as a function of the total concentration of unlabeled ProTα, $c_{\text{P}}^{\text{tot}}$.

The mean dwell time of the low-FRET state is given by $\tau_{\text{low}}^{\text{P}} = \left(k_{\text{on}}c_{\text{H}} + k_{\text{on}}^{\text{PPH}}c_{\text{PH}}\right)^{-1}$. An analytical expression for $\tau_{\text{high}}^{\text{P}}$ is not easily found for the four-state system. However, under the conditions studied here, with $c_{\text{H}}^{\text{tot}} = 10$ nM, the population of P*HH is negligible owing to the low affinity of H to PH ($K_{\text{D}} = 12 \pm 3$ μM) and the excess of P. With this assumption, we can reduce the problem to a three-state system with states P*,

P*H, P*PH, and the rate matrix

$$\mathbf{K}_{3,\text{P}} = \begin{pmatrix} -k_{\text{on}}c_{\text{H}} - k_{\text{on}}^{\text{PPH}}c_{\text{PH}} & k_{\text{off}} & k_{\text{off}}^{\text{PPH}}/2 \\ k_{\text{on}}c_{\text{H}} & -k_{\text{off}} - k_{\text{on}}^{\text{PPH}}c_{\text{P}} & k_{\text{off}}^{\text{PPH}}/2 \\ k_{\text{on}}^{\text{PPH}}c_{\text{PH}} & k_{\text{on}}^{\text{PPH}}c_{\text{P}} & -k_{\text{off}}^{\text{PPH}} \end{pmatrix}, \quad (14)$$

for which we find

$$\tau_{\text{high}}^{\text{P}} = \frac{2c_{\text{H}}k_{\text{on}}(k_{\text{off}}^{\text{PPH}} + c_{\text{P}}k_{\text{on}}^{\text{PPH}}) + c_{\text{PH}}k_{\text{on}}^{\text{PPH}}(2k_{\text{off}} + k_{\text{off}}^{\text{PPH}} + 2c_{\text{P}}k_{\text{on}}^{\text{PPH}})}{k_{\text{off}}^{\text{PPH}}(2k_{\text{off}} + c_{\text{P}}k_{\text{on}}^{\text{PPH}})(c_{\text{H}}k_{\text{on}} + c_{\text{PH}}k_{\text{on}}^{\text{PPH}})}. \quad (15)$$

$k_{\text{on}}$ and $k_{\text{off}}$ were determined independently from measurements in the absence of unlabeled ProTα ($c_{\text{P}}^{\text{tot}} = 0$) (Fig. 3b). The concentrations of unbound P, unbound H, and PH ($c_{\text{P}}$, $c_{\text{H}}$, $c_{\text{PH}}$) can be related to the total concentrations based on Eq. (12). We observed apparent two-state behavior between a low-FRET and a high-FRET state under all conditions. From the model, the mean dwell times of the low and the high-FRET states were calculated, and from the corresponding inverse expressions, the equation describing the observed pseudo-first-order rates coefficients $k_{\text{on}}^{\text{obs,P}}$ and $k_{\text{off}}^{\text{obs,P}}$ as functions of all the individual rate coefficients and the concentrations of H1 and ProTα. The resulting functions were used to globally fit the results obtained from all kinetic experiments. As a result, $k_{\text{on}}^{\text{PPH}}$ and $k_{\text{off}}^{\text{PPH}}$ were the only adjustable parameters, which were shared in the global fit of all kinetic data (Fig. 3).

To compare the exchange rates of ProTα and H1, defined as $k_{\text{ex}}^{\text{P,4-state}} = k_{\text{on}}^{\text{obs,P}} + k_{\text{off}}^{\text{obs,P}}$ and $k_{\text{ex}}^{\text{H,4-state}} = k_{\text{on}}^{\text{obs,H}} + k_{\text{off}}^{\text{obs,H}}$, respectively, under different experimental conditions (Supplementary Fig. 2), we numerically calculated the observed pseudo-first-order rates $k_{\text{on}}^{\text{obs,P}} = 1/\langle\tau_{\text{low}}^{\text{P}}\rangle$, $k_{\text{off}}^{\text{obs,P}} = 1/\langle\tau_{\text{high}}^{\text{P}}\rangle$, $k_{\text{on}}^{\text{obs,H}} = 1/\langle\tau_{\text{low}}^{\text{H}}\rangle$, and $k_{\text{off}}^{\text{obs,H}} = 1/\langle\tau_{\text{high}}^{\text{H}}\rangle$ derived from the rate matrices of the 4-state model, $\mathbf{K}_{4,\text{P}}$ (Eq. 13) and

$$\mathbf{K}_{4,\text{H}} = \begin{pmatrix} -k_{\text{on}}c_{\text{P}} - k_{\text{on}}^{\text{PHH}}c_{\text{PH}} & k_{\text{off}} & 0 & k_{\text{off}}^{\text{PHH}}/2 \\ k_{\text{on}}c_{\text{P}} & -k_{\text{off}} - k_{\text{on}}^{\text{PPH}}c_{\text{P}} - k_{\text{on}}^{\text{PHH}}c_{\text{H}} & k_{\text{off}}^{\text{PPH}} & k_{\text{off}}^{\text{PHH}}/2 \\ 0 & k_{\text{on}}^{\text{PPH}}c_{\text{P}} & -k_{\text{off}}^{\text{PPH}} & 0 \\ k_{\text{on}}^{\text{PHH}}c_{\text{PH}} & k_{\text{on}}^{\text{PHH}}c_{\text{H}} & 0 & -k_{\text{off}}^{\text{PHH}} \end{pmatrix}, \quad (16)$$

which is valid for the observation of labeled H (i.e., for experiments where H is fluorescence- or isotope-labeled). The four states are H*, P*, PPH*, and PH*H. For the 2-state model, we used:

$$k_{\text{ex}}^{\text{P,2-state}} = k_{\text{on}}c_{\text{H}} + k_{\text{off}}$$
$$k_{\text{ex}}^{\text{H,2-state}} = k_{\text{on}}c_{\text{P}} + k_{\text{off}} \quad . \quad (17)$$

**NMR spectroscopy.** All NMR data were recorded in TBS buffer (with an ionic strength of 200 mM adjusted with KCl), pH 7.4, 10% D$_2$O (v/v), and 0.7 mM 4,4-dimethyl-4-silapentane-1-sulfonic acid (DSS). NMR spectra were acquired at 283 K on a Bruker AVANCE III 750 MHz (¹H) spectrometer equipped with a cryogenic probe. Free induction decays (FIDs) were transformed and visualized in NMRPipe[57] or Dynamics Center (Bruker Biospin) and analyzed using the CcpNmr Analysis software[58] or Topspin (Bruker Biospin). Proton chemical shifts were referenced internally to DSS at 0.00 ppm, with heteronuclei referenced by relative gyromagnetic ratios.

¹H,¹⁵N-HSQC spectra of ¹⁵N-labeled ProTα (20 μM) were recorded in the absence and presence of different concentrations of H1 (0–80 μM). Before titration experiments, the proteins were dialyzed in the same beaker. Subsequently, the solution of labeled protein was split equally into two samples, to one of which the unlabeled titrant was added at the maximum concentration, and to the other the same volume of dialysis buffer. After the acquisition of NMR spectra on the two samples, they were used to obtain titration points between the endpoints by sequentially mixing the sample of the complex with the unbound protein.

Values of $R_{\text{H}}$ from NMR of 20 μM ¹⁵N-labeled ProTα alone and with different concentrations of H1 were determined from a series of ¹H,¹⁵N-HSQC spectra with preceding pulse-field gradient stimulated-echo longitudinal encode-decode (PG-SLED) diffusion filter[59] and with the gradient strength increasing linearly from 0.963 to 47.2 G cm$^{-1}$. To determine the diffusion coefficients, $D$, the decay curves of the amide peaks were plotted against the gradient strength and fitted in Dynamics Center (Bruker Biospin) with the Stejskal–Tanner equation, $I = I_0 \exp\left[-D\,G_{\text{x}}^2\gamma^2\delta^2\left(\Delta - \frac{\delta}{3}\right)\right]$, with $I$ being the intensity of the NMR signal at the respective gradient strength, $I_0$ the intensity without applied gradient, $G_{\text{x}}$ the gradient strength, $\gamma = 26752$ rad G$^{-1}$ s$^{-1}$ the gyromagnetic ratio, $\delta = 3$ ms the gradient pulse width, and $\Delta = 250$ ms the pulse separation, or diffusion time. All samples contained 0.05% (v/v) 1,4-dioxane as an internal viscosity reference. $R_{\text{H}}$ was calculated from the relative diffusion decays of ProTα and 1,4-dioxane, which has an $R_{\text{H}}$ of 2.12 Å[60], using $R_{\text{H}}^{\text{protein}} = D^{\text{dioxane}}R_{\text{H}}^{\text{dioxane}}/D^{\text{protein}}$, where $D^{\text{protein}}$ and $D^{\text{dioxane}}$ are the measured diffusion coefficients.

**NMR lineshape calculations.** Quantifying the rate coefficients of the kinetic model directly from lineshape analysis by fitting the NMR spectra is complicated

by the complexity of the kinetic model and the relatively large number of parameters, including the chemical shifts of the ternary complexes, which are difficult to determine independently. We thus compared the measured 1D-$^{15}$N lineshapes in a titration of 20 μM $^{15}$N-ProTα with unlabeled H1 to the lineshapes calculated using the Bloch–McConnell equations[26,61] adapted for multi-state kinetics using the kinetic model underlying the ProTα-H1 interaction. We write $M_{xy}^{j,s} = M_x^{j,s} + iM_y^{j,s}$ for the transverse nuclear magnetization of residue $j$ of the subpopulation in chemical state $s$. Here we consider four states, $s = P$, PH, PPH, or PHH. With these four magnetizations combined in the vector $\mathbf{M}^j = \left(M_{xy}^{j,P}, M_{xy}^{j,PH}, M_{xy}^{j,PPH}, M_{xy}^{j,PHH}\right)^{\mathrm{T}}$, the Bloch–McConnell equation that describes an FID in a static magnetic field of magnitude $B_0$ reads:

$$\frac{d\mathbf{M}^j}{dt} = \left(-i\mathbf{\Omega}^j + \mathbf{R}_2^j + \mathbf{K}_4\right)\mathbf{M}^j, \tag{18}$$

where $\mathbf{\Omega}^j$ and $\mathbf{R}_2^j$ are diagonal matrices with the Larmor frequencies $\omega_0^{j,s} = \gamma_{j,s}B_0$ and the relaxation rates $R_2^{j,s}$ of the four subpopulations as diagonal elements, respectively. The solution of Eq. (18) is given using the matrix exponential function:

$$\mathbf{M}^j(t) = e^{\left(-i\mathbf{\Omega}^j + \mathbf{R}_2^j + \mathbf{K}_4\right)t}\mathbf{M}_0^j, \tag{19}$$

where $\mathbf{M}_0^j = M_0^j\mathbf{p}_{\mathrm{eq}}$ is the transverse magnetization of the FID at $t = 0$. $\mathbf{p}_{\mathrm{eq}}$ is the equilibrium distribution given by $\mathbf{K}_4\mathbf{p}_{\mathrm{eq}} = 0$ and $\mathbf{1}^{\mathrm{T}}\mathbf{p}_{\mathrm{eq}} = 1$. The contribution of residue $j$ to the measured signal is then proportional to $s_j(t) = \mathbf{1}^{\mathrm{T}}\mathbf{M}^j(t)$. Fourier transforming this result yields the frequency-domain signal:

$$S_j(\omega) = \int_0^\infty s_j(t)e^{i\omega t}dt, \tag{20}$$

of which we take the real part.

$\mathbf{K}_{4,\mathrm{P}}$ (Eq. (13)) was determined using the rate coefficients from the global analysis of the fluorescence kinetics (Fig. 4) and from the H1 titration of labeled ProTα, as shown in Supplementary Fig. 4. $\mathbf{\Omega}^j$ and $\mathbf{R}_2^j$ can be quantified from the positions and widths, respectively, of the NMR resonances. We focus our analysis on the $^{15}$N dimension, which exhibits the dominant chemical shift changes. We used the 1D projections of ProTα peaks with the largest chemical shift changes and sufficiently free from overlap with other resonances on the $^{15}$N frequency axis. The projections at different concentrations of H1 were normalized by the peak volume for each spectrum after background correction, and fit with Lorentzian peak functions to extract the peak position $\omega_0^j(c_{\mathrm{H}}^{\mathrm{tot}})$ and the full width at half height $\Delta\omega_{FWHH}^j(c_{\mathrm{H}}^{\mathrm{tot}})$ (both in units of rad s$^{-1}$). $\omega_0^{j,\mathrm{P}}$ is identical to $\omega_0^j(c_{\mathrm{H}}^{\mathrm{tot}} = 0)$. The other values of $\mathbf{\Omega}^j$ were obtained by minimizing

$$\chi^2 = \sum_{c_{\mathrm{H}}^{\mathrm{tot}}} \left[\omega_0^j(c_{\mathrm{H}}^{\mathrm{tot}}) - \mathbf{1}^{\mathrm{T}}\mathbf{\Omega}^j\mathbf{p}_{\mathrm{eq}}(c_{\mathrm{H}}^{\mathrm{tot}})\right]^2, \tag{21}$$

with respect to $\omega_0^{j,\mathrm{PH}}$, $\omega_0^{j,\mathrm{PPH}}$, and $\omega_0^{j,\mathrm{PHH}}$, summing over all $c_{\mathrm{H}}^{\mathrm{tot}}$ at which $\omega_0^j(c_{\mathrm{H}}^{\mathrm{tot}})$ was measured. We calculated the apparent transverse relaxation rates as $R_2^{j,\mathrm{P}} = \frac{1}{2}\Delta\omega_{FWHH}^j(c_{\mathrm{H}}^{\mathrm{tot}} = 0)$, and we assume $R_2^{j,\mathrm{PH}} = R_2^{j,\mathrm{PPH}} = R_2^{j,\mathrm{PHH}} = \frac{1}{2}\Delta\omega_{FWHH}^j(20\,\mu\mathrm{M})$. The latter approximation is justified by the small changes in line width above 20 μM H1 (Supplementary Fig. 6f), an observation that is not unexpected for IDPs, where local backbone dynamics can dominate orientational decorrelation of the NH vectors[62], especially in an extremely disordered and dynamic system such as H1-ProTα. We tested the robustness of the lineshape calculation by systematic parameter variation within reasonable bounds (Supplementary Fig. 6).

**Simulations.** A coarse-grained simulation model was used to model the association of ProTα-H1 complexes in different stoichiometries. The model was identical to that used in our previous work for intramolecular energy terms and for ProTα–H1 interactions[12]. The interactions between two ProTα chains or between two H1 chains also used the same energy function, but native interactions within the H1 globular domain were only applied to native residue pairs within the same chain; the non-native interaction energy term was instead applied between these pairs if the two residues were in different chains. A Debye length of 0.69 nm was used, reflecting the 200 mM ionic strength in the experiments. Simulations were run with Gromacs[63] version 4.0.5 or 5.1.4, using periodic boundary conditions with a cubic cell of edge 80 nm, and a 2.5-nm cutoff was applied to non-bonded interactions. Dynamics were propagated via a Langevin algorithm with a friction coefficient of 0.2 ps$^{-1}$ and a time step of 10 fs at 300 K.

The potential of mean force (PMF), $W(r)$, was determined between centers of mass using umbrella sampling with 32 replicas in each case, with minima at 0, 0.5, 1.0, 1.5, 2., 2.5, 3., 3.5, 4., 4.5, 5.5, 6.5, 7.5, 8.5, 9.5, 10.5, 11.5, 12.5, 13.5, 14.5, 15.5, 16.5, 17.5, 18.5, 19.5, 20.5, 21.5, 22.5, 23.5, 24.5, 25.5, and 26.5 nm, and force constants of 10 kJ mol$^{-1}$ nm$^{-2}$. Weighted histogram analysis[64] was used to determine $W(r)$, from which the effective two-body potential $F(r)$ was obtained as

$F(r) = W(r) + 2k_BT\ln r$. Values of $K_D$ were determined from $F(r)$ via $K_D^{-1} = 4\pi N_A\int_0^c\exp[-\beta F(r)]r^2dr$, where $N_A$ is the Avogadro constant, and $c$ is the radius above which $F(r)$ is zero. PMFs were determined for the association of ProTα with H1, or with existing ProTα-H1 complexes, and similarly for H1 with existing ProTα-H1 complexes. In each case, the umbrella potentials were applied between the centers of mass of the two species being separated (either single proteins or complexes). Association rate coefficients, $k_{\mathrm{on}}$, were calculated from the PMFs using

$$N_A k_{\mathrm{on}}^{-1} = \int_b^c \frac{\exp[\beta F(r)]}{4\pi r^2 D}\,dr + (4\pi Dc)^{-1}, \tag{22}$$

which gives the reactive flux onto an absorbing boundary at radius $b$[65]. The constant diffusion coefficient, $D$, was taken as the sum of the experimentally measured translational diffusion coefficients of the associating species. A "capture radius" can be defined as the largest radius at which the calculated $k_{\mathrm{on}}$ becomes constant (Supplementary Fig. 7c), indicating a bound state.

**Reporting summary**. Further information on research design is available in the Nature Research Reporting Summary linked to this article.

## Data availability
Data supporting the findings of this paper are available from the corresponding authors upon reasonable request. A reporting summary for this article is available as a Supplementary Information file. Source data are provided with this paper.

## Code availability
A custom WSTP add-on for Mathematica (Wolfram Research) used for the analysis of single-molecule fluorescence data is available upon request and at https://schuler.bioc.uzh.ch/programs/.

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

## Acknowledgements

We thank Erik Holmstrom, Louise Pinet, Andreas Prestel, and Chris Waudby for helpful discussions, Iwo König and Jacob Hertz Martinsen for protein production, Jendrik Schöppe and Andreas Plückthun for the pAT222-pD expression vector, and the Functional Genomics Center Zurich for performing mass spectrometry. This work was supported by the Swiss National Science Foundation (to B.S.), the Forschungskredit of the University of Zurich (to A.S., FK-17-038, and A.C., FK-19-039), the Novo Nordisk Foundation Challenge program REPIN (NNF18OC0033926 to B.B.K. and B.S.), and the Intramural Research Program of the NIDDK at the National Institutes of Health (R.B.B.). This work utilized the computational resources of the NIH HPC Biowulf cluster (http://hpc.nih.gov).

## Author contributions

A.S., R.B., B.B.K., and B.S. designed the research; A.S., A.B., K.B., D.N., A.C., R.B., B.B.K., and B.S. performed the research; A.S., A.B., K.B., M.B., D.N., P.H., and F.Z. contributed new reagents or analytical tools; A.S., A.B., K.B., A.C., P.H., D.N., and R.B. analyzed the data; A.S., D.N., and B.S. wrote the paper with contributions from all authors.

## Competing interests

The authors declare no competing interests.
