## [Peer Review File · Nature Communications]

REVIEWERS' COMMENTS:

Reviewer #1 (Remarks to the Author):

This manuscript studies complex formation of two highly charged intrinsically disordered proteins (the linker histone H1.0 and prothymosin α) that maintain their disordered and fast relaxation dynamics in the bound state. The manuscript describes the association/dissociation kinetics over a broad range of protein concentrations (0.05 to 10 μ M) for both proteins. They observe that the kinetics switch from slow (two-state) to fast (non-two state) at higher concentrations. The switch in kinetics is described by coarse grained simulations that provide a mechanistic description (similar to the fly-casting mechanism proposed by Wolynes' group [ref 28]).

This manuscript masterfully combines single-molecule FRET, NMR, computation, creative combination of labeled/unlabeled protein titrations and rigorous analysis of all the data to describe the high affinity binding of two, highly charged, intrinsically disordered protein to form 1:1 dimers and ternary complexes over a broad range of concentrations of both components from picomolar to nanomolar concentrations.

The key (most beautiful) evidence is provide in Fig. 4. In particular the global fit of the kinetic data from different methods as a function of concentration. Using the notation H= histone H1.0 and P= prothymosin α , they studied the kinetics of $H+P \rightarrow PH$ and $PH + P \rightarrow PPH$. In this analysis they find that the dissociation is almost three orders of magnitude faster for the PPH trimer than for the PH dimer. Coarse grained MD simulations show that the mechanism of this fast exchange depends on three things: strong, non-specific association (i.e., two polyelectrolytes interacting), fast relaxation (100 ns) and maintenance of the disorder in the bound states. These three conditions also facilitates the modeling, since simple CG description of the system can capture the dynamics of the system.

The simulations show that the fast chain exchange mechanism can be explained by the formation of ternary complex. A 1:1 complex is formed and equilibrates quickly and a transient 2:1 ternary intermediate is formed. The disorder of the complex (and the incoming chain) allows for favorable interactions at distances that are much larger than the hydration radius of the molecules e.g., flycasting) thus facilitating the rapid exchange of either one of the chains in the 2:1 intermediate.

The manuscript is well written. All methods are carefully described. The results are novel. The work is highly relevant and adds new insights into the role of intrinsically disorder proteins in the regulation of protein interactions over a broad range of protein concentrations. I recommend publication.

Reviewer #2 (Remarks to the Author):

In this manuscript, Sottini and colleagues expand upon their prior characterization of complex formation between two oppositely charged intrinsically disordered proteins, ProT α and H1. Here, the focus is on the potential for formation of higher order complexes beyond 1:1 stoichiometry that result in concentration dependent dissociation kinetics. The authors convincingly demonstrate that the seemingly discrepant exchange behavior observed at low protein concentrations (slow exchange by single-molecule spectroscopy) and high protein concentrations (fast exchange by NMR spectroscopy) can best be explained by a unified kinetic mechanism that involves formation of transient ternary complexes, which would be favored at high protein concentrations. The integrated approach of single-molecule fluorescence, NMR, and molecular simulations used here is well-suited to addressing the scientific questions at hand and the facilitated dissociation mechanism described in this work should be of great interest to the readers of Nature Communications and the scientific community at large.

Minor suggestions that should improve the clarity of manuscript are included below:

- The data in Figure 4a deviate from the model at delay times > 1 s. Why? Is this due to the occurrence of new molecules in the focal volume?
- In the lineshape analysis of the NMR titration data, it seems that the authors have made a simplifying assumption that the PH, PPH, and PHH states would have the same R_2 (see legend of Supplementary Figure 6). This seems unlikely given the increase in molecular size of the ternary complexes, which one would expect to be highly populated at H1 concentrations > 20 μM , and the dominant species at $[\text{H1}] > 50$ μM , based on the calculations presented in Supplementary Figure 3h. Additionally, while the 1D 15N projections of NMR data are in reasonable agreement with the calculated 4-state behavior, there are some notable differences, particularly at higher H1 concentrations (although this is sometimes hard to decipher due to the limited color scheme used for these figures). Some discussion of this would be extremely helpful to reconcile these apparent inconsistencies, if they do exist.

Reviewer #3 (Remarks to the Author):

In this manuscript, Sottini et al. present a very solid and exceptional study on the concentration dependent association and dissociation kinetics of two disordered proteins prothymosin alpha (ProTa, P) and linker histone H1.0 (H1, H) using a combination of single molecule FRET spectroscopy, NMR, and molecular simulation. Both ProTa and H1 are disordered and highly, oppositely charged (net charges of -44 and +53, respectively) and the authors have shown that the two proteins form a dynamic complex with a surprisingly high binding affinity (picomolar dissociation constant), but without structure formation at low concentrations. Now the authors report another surprising finding that the binding kinetics becomes much faster at higher, physiological protein concentrations. They employed various advanced single-molecule FRET techniques developed in the Schuler group, NMR, and coarse-grained molecular dynamics simulations to show this strange concentration-dependent kinetics results from the formation of ternary complexes PPH and PHH. The dissociation of the second P (or H) molecule is several orders of magnitude faster than the first one, which results in very different kinetics at low and high protein concentrations. Very clean and high quality experimental data, sophisticated data analyses and interpretations based on the simulation results (such as different ionic strength-dependence of the association and dissociation rates) convincingly support the ternary complex model. I strongly recommend publication of this work and hope the authors can address only several minor points listed below.

1. Fig. 4a caption: "Dotted lines show the contribution 283 to the relaxation due to ..."
I suppose "dotted line" means "dashed line"
2. Fig. 2h caption: It becomes clear by reading the later part of the manuscript, but it may be a good idea to mention "4 states" explicitly in the figure caption for easier understanding.
3. In Fig. 2a - c, the FRET efficiency in the absence of unlabeled ProTa is slightly lower (by ~ 0.05) for 20 micromolar H1 compared to those at 10 nM and 1 micromolar. Is this due to the formation of the ternary complex PHH? Does the binding of two H1 molecules simply extend ProTa?
4. I have questions on the validity of the rate matrices in equations (13) - (16). First, since the reactions are not linear, it is not clear if the rate matrix approach would work. If it works, the complete matrix will be as follows, which consists of five states: P, H, PH, PPH, PHH.

(see attached file for the equation)

The authors used reduced matrices by omitting H for eq 13, H and PHH for eq 14, and P for eq 16. I am not sure if this approach will give the same results as that using the complete matrix above. In

addition, I don't understand why $k_{\text{off}}^{\text{PPH}}$ was divided by 2 in the first and second rows in eq (13) (same for $k_{\text{off}}^{\text{PHH}}$ in eq (16)). Since the dissociation of PH, PPH, and PHH will produce two molecules, the sum of columns would not necessarily be equal to 0. Dividing the rate constant by 2 in eq (14) may be correct. There are two dissociation pathways for P^*PH because P^* and P are distinguishable due to partial labeling in this case and only one pathway ($\text{P}^*\text{PH} \rightarrow \text{P}^*\text{H} + \text{P}$ or $\text{P}^*\text{PH} \rightarrow \text{P}^* + \text{PH}$) contributes to the production of P^*H or P^* . However, in this equation (14), there would be more states, P, P^* , PH, and P^*H instead of P^* and P^*H , which makes the rate matrix more complex. I hope the authors comment on these points.

Sottini et al.

RESPONSE TO REVIEWERS' COMMENTS:

Reviewer #1 (Remarks to the Author):

This manuscript studies complex formation of two highly charged intrinsically disordered proteins (the linker histone H1.0 and prothymosin α) that maintain their disordered and fast relaxation dynamics in the bound state. The manuscript describes the association/dissociation kinetics over a broad range of protein concentrations (0.05 to 10 μ M) for both proteins. They observe that the kinetics switch from slow (two-state) to fast (non-two state) at higher concentrations. The switch in kinetics is described by coarse grained simulations that provide a mechanistic description (similar to the fly-casting mechanism proposed by Wolynes' group [ref 28]).

This manuscript masterfully combines single-molecule FRET, NMR, computation, creative combination of labeled/unlabeled protein titrations and rigorous analysis of all the data to describe the high affinity binding of two, highly charged, intrinsically disordered protein to form 1:1 dimers and ternary complexes over a broad range of concentrations of both components from picomolar to nanomolar concentrations.

The key (most beautiful) evidence is provide in Fig. 4. In particular the global fit of the kinetic data from different methods as a function of concentration. Using the notation H= histone H1.0 and P= prothymosin α , they studied the kinetics of $H+P \rightarrow PH$ and $PH+P \rightarrow PPH$. In this analysis they find that the dissociation is almost three orders of magnitude faster for the PPH trimer than for the PH dimer. Coarse grained MD simulations show that the mechanism of this fast exchange depends on three things: strong, non-specific association (i.e., two polyelectrolytes interacting), fast relaxation (100 ns) and maintenance of the disorder in the bound states. These three conditions also facilitates the modeling, since simple CG description of the system can capture the dynamics of the system.

The simulations show that the fast chain exchange mechanism can be explained by the formation of ternary complex. A 1:1 complex is formed and equilibrates quickly and a transient 2:1 ternary intermediate is formed. The disorder of the complex (and the incoming chain) allows for favorable interactions at distances that are much larger than the hydration radius of the molecules e.g., flycasting) thus facilitating the rapid exchange of either one of the chains in the 2:1 intermediate.

The manuscript is well written. All methods are carefully described. The results are novel. The work is highly relevant and adds new insights into the role of intrinsically disorder proteins in the regulation of protein interactions over a broad range of protein concentrations. I recommend publication.

We thank the reviewer for the enthusiastic assessment of our work.

Reviewer #2 (Remarks to the Author):

In this manuscript, Sottini and colleagues expand upon their prior characterization of complex formation between two oppositely charged intrinsically disordered proteins, ProT α and H1. Here, the focus is on the potential for formation of higher order complexes beyond 1:1 stoichiometry that result in concentration dependent dissociation kinetics. The authors convincingly demonstrate that the seemingly discrepant exchange behavior observed at low protein concentrations (slow exchange by single-molecule spectroscopy) and high protein concentrations (fast exchange by NMR

spectroscopy) can best be explained by a unified kinetic mechanism that involves formation of transient ternary complexes, which would be favored at high protein concentrations. The integrated approach of single-molecule fluorescence, NMR, and molecular simulations used here is well-suited to addressing the scientific questions at hand and the facilitated dissociation mechanism described in this work should be of great interest to the readers of Nature Communications and the scientific community at large.

We thank the reviewer for the very positive evaluation of our work and the suggestions for improvements, which we address in detail below.

Minor suggestions that should improve the clarity of manuscript are included below:

- The data in Figure 4a deviate from the model at delay times > 1 s. Why? Is this due to the occurrence of new molecules in the focal volume?

The range between about 1 and 10 ms, where we observe some deviation from the fit, is indeed most strongly affected by the occurrence of new molecules, as shown by the dashed lines. Contributions in this range may also arise from residual fluorescent impurities that do not participate in binding reactions or by photobleaching.

- In the lineshape analysis of the NMR titration data, it seems that the authors have made a simplifying assumption that the PH, PPH, and PHH states would have the same R_2 (see legend of Supplementary Figure 6). This seems unlikely given the increase in molecular size of the ternary complexes, which one would expect to be highly populated at H1 concentrations > 20 μM , and the dominant species at $[\text{H1}] > 50$ μM , based on the calculations presented in Supplementary Figure 3h.

Thank you for pointing out that we may not have made this point sufficiently clear. In the line shape calculations, we have indeed made the approximation that the values of R_2 of ProT α in PH, PPH, and PHH are the same. Strictly speaking, the reviewer is correct, and different values might be expected for the different oligomeric species. However, a comparison of the linewidths at 20 μM H1 (where PH is maximally populated) and 80 μM H1 (where PHH dominates, Supplementary Fig. 6f) shows only small changes close to the scatter of the data, and no systematic trend towards lower or higher values for the different resonances, indicating that R_2 in this system does not exhibit a simple relation with the size of the complex. This behavior is not unexpected for IDPs, where local backbone dynamics can dominate orientational decorrelation of the NH vectors (see, e.g., Konrat *J. Magn. Reson.* 241, 74-85), especially in an extremely disordered and dynamic system as H1-ProT α . In the absence of a uniform trend of linewidth with the size of the complex, the linewidths for both PHH and PPH would essentially have to be treated as additional residue-specific fit parameters, leading to a large increase in the number of free parameters in the model. We now included an explicit statement regarding these considerations for our assumption regarding R_2 in the Methods section and the above reference.

Additionally, while the 1D ^{15}N projections of NMR data are in reasonable agreement with the calculated 4-state behavior, there are some notable differences, particularly at higher H1 concentrations (although this is sometimes hard to decipher due to the limited color scheme used for these figures). Some discussion of this would be extremely helpful to reconcile these apparent inconsistencies, if they do exist.

We are not entirely sure which discrepancies the reviewer is referring to. Given the signal-to-noise ratio at the relatively low concentration of 20 μM ^{15}N ProT α and the additional effects from background subtraction and residual signal overlap with neighboring resonances on peak integration

and normalization, we consider the agreement to be quite good. For illustration, we include direct overlays between the data points and the calculated peaks below.

Reviewer #3 (Remarks to the Author):

In this manuscript, Sottini et al. present a very solid and exceptional study on the concentration dependent association and dissociation kinetics of two disordered proteins prothymosin alpha (ProTa, P) and linker histone H1.0 (H1, H) using a combination of single molecule FRET spectroscopy, NMR, and molecular simulation. Both ProTa and H1 are disordered and highly, oppositely charged (net charges of -44 and +53, respectively) and the authors have shown that the two proteins form a dynamic complex with a surprisingly high binding affinity (picomolar dissociation constant), but without structure formation at low concentrations. Now the authors report another surprising finding that the binding kinetics becomes much faster at higher,

physiological protein concentrations. They employed various advanced single-molecule FRET techniques developed in the Schuler group, NMR, and coarse-grained molecular dynamics simulations to show this strange concentration-dependent kinetics results from the formation of ternary complexes PPH and PHH. The dissociation of the second P (or H) molecule is several orders of magnitude faster than the first one, which results in very different kinetics at low and high protein concentrations. Very clean and high quality experimental data, sophisticated data analyses and interpretations based on the simulation results (such as different ionic strength-dependence of the association and dissociation rates) convincingly support the ternary complex model. I strongly recommend publication of this work and hope the authors can address only several minor points listed below.

We thank the reviewer for the appreciation of our results and the helpful comments, which we address below.

1. Fig. 4a caption: “Dotted lines show the contribution 283 to the relaxation due to ...”
I suppose “dotted line” means “dashed line”

Thanks for noticing this discrepancy, which we have now corrected. To eliminate possible confusion, we also deleted the horizontal black line, which is dispensable.

2. Fig. 2h caption: It becomes clear by reading the later part of the manuscript, but it may be a good idea to mention “4 states” explicitly in the figure caption for easier understanding.

We do in fact mention the four-state model both in the title of the panel and in the caption of Fig. 2: “(g,h) Comparison of NMR 1D ¹⁵N lineshapes calculated using the Bloch-McConnell equation for a two-state (g) or a four-state binding mechanism (h) (see Supplementary Table 2, for rate coefficients, and Methods for details).”

3. In Fig. 2a - c, the FRET efficiency in the absence of unlabeled ProTa is slightly lower (by ~ 0.05) for 20 micromolar H1 compared to those at 10 nM and 1 micromolar. Is this due to the formation of the ternary complex PHH? Does the binding of two H1 molecules simply extend ProTa?

The transfer efficiency of double-labeled ProTα bound to H1 indeed decreases at high concentrations of H1. This behavior is most clearly seen in Supplementary Figure 3d, and as the reviewer suggests, we interpret it as the formation of higher-order complexes. Since this connection may not be obvious, we now included a reference to Supplementary Figure 3d in the caption of Fig. 2. Thank you for pointing it out!

4. I have questions on the validity of the rate matrices in equations (13) - (16). First, since the reactions are not linear, it is not clear if the rate matrix approach would work. If it works, the complete matrix will be as follows, which consists of five states: P, H, PH, PPH, PHH.

$$\begin{pmatrix} -k_{on}^{CH} - k_{on}^{PPH} C_{PH} & 0 & k_{off} & k_{off}^{PPH} & 0 \\ 0 & -k_{on}^{CP} - k_{on}^{PHH} C_{PH} & k_{off} & 0 & k_{off}^{PHH} \\ k_{on}^{CH} & k_{on}^{CP} & -k_{off} - k_{on}^{PPH} C_{P} - k_{on}^{PHH} C_{H} & k_{off}^{PPH} & k_{off}^{PHH} \\ k_{on}^{PPH} C_{PH} & 0 & k_{on}^{PPH} C_{P} & -k_{off}^{PPH} & 0 \\ 0 & k_{on}^{PHH} C_{PH} & k_{on}^{PHH} C_{H} & 0 & -k_{off}^{PHH} \end{pmatrix} \begin{pmatrix} P \\ H \\ PH \\ PPH \\ PHH \end{pmatrix}$$

The authors used reduced matrices by omitting H for eq 13, H and PHH for eq 14, and P for eq 16.

I am not sure if this approach will give the same results as that using the complete matrix above. In addition, I don't understand why $k_{\text{off}}^{\text{PPH}}$ was divided by 2 in the first and second rows in eq (13) (same for $k_{\text{off}}^{\text{PHH}}$ in eq (16)). Since the dissociation of PH, PPH, and PHH will produce two molecules, the sum of columns would not necessarily be equal to 0. Dividing the rate constant by 2 in eq (14) may be correct. There are two dissociation pathways for P^*PH because P^* and P are distinguishable due to partial labeling in this case and only one pathway ($\text{P}^*\text{PH} \rightarrow \text{P}^*\text{H} + \text{P}$ or $\text{P}^*\text{PH} \rightarrow \text{P}^* + \text{PH}$) contributes to the production of P^*H or P^* . However, in this equation (14), there would be more states, P , P^* , PH , and P^*H instead of P^* and P^*H , which makes the rate matrix more complex. I hope the authors comment on these points.

We regret that we may have caused some of the possible confusion, which we thank the reviewer for pointing out, by stating above Eq. 13: "ProT α can thus populate four states, P , PH , PPH , and PHH ; ...". We now replaced this statement by the more specific "Labeled ProT α (P^*) can thus populate four states, P^* , P^*H , P^*PH , and P^*HH ; the formation of $\text{P}^*\text{P}^*\text{H}$ is negligible because $c_{\text{P}^*}^{\text{tot}} \ll c_{\text{P}}^{\text{tot}}$...".

We agree with Reviewer #3 that the kinetics described in Eq. 12 are nonlinear in the concentrations, and the rate matrix approach does not work in that case, since the matrix itself would depend on the concentrations, and the rate equations cannot be solved by a simple matrix exponential. However, it is important to note that with the matrix of Eq. 13, we aim to describe only the kinetics of complexes containing labeled ProT α (P^*), i.e. those complexes which are observed in the experiments. P^* is present in exceedingly small amounts in the samples compared to unlabeled ProT α (P). Consequently, the formation of $\text{P}^*\text{P}^*\text{H}$ is negligible, and labeled ProT α is observed in only four states, P^* , P^*H , P^*PH , and P^*HH .

Another consequence of $c_{\text{P}^*}^{\text{tot}} \ll c_{\text{P}}^{\text{tot}}$ in our measurements is that the addition of P^* does not change the equilibrium concentrations c_{H} , c_{P} , c_{PH} , c_{PPH} , and c_{PHH} to a detectable extent. These concentrations are therefore only dependent on the equilibrium constants of the reactions of Eq. 12 and the total concentrations $c_{\text{P}}^{\text{tot}}$ and $c_{\text{H}}^{\text{tot}}$, and we can describe the populations of the four states by a set of four linear rate equations (linear in the concentrations c_{P^*} , $c_{\text{P}^*\text{H}}$, $c_{\text{P}^*\text{PH}}$, and $c_{\text{P}^*\text{HH}}$). Note that the corresponding rate matrix is independent of these concentrations; it only depends on the concentrations of complexes with unlabeled ProT α (c_{P} , c_{PH} , c_{PPH} , and c_{PHH}) and on c_{H} . We added a sentence to the manuscript to make this point clearer.

Further, regarding the factor of 2: P^*PH dissociates either to $\text{P}^* + \text{PH}$ or to $\text{P} + \text{P}^*\text{H}$, as the reviewer points out. We assume here that both processes are equally likely; in the four-state system, which tracks only the labeled species, the depopulation of P^*PH thus leads to a population of the states P^* and P^*H , both with rates $\frac{1}{2}k_{\text{off}}^{\text{PPH}}c_{\text{P}^*\text{PH}}$, where $k_{\text{off}}^{\text{PPH}}$ is the same as in Eq. 12; however, the two ProT α are considered indistinguishable. The same line of reasoning described above applies to Eq. 16, where kinetics of complexes with labeled histone is described by the four states H^* , PH^* , PPH^* , and PH^*H , and by a rate matrix that depends solely on the concentrations of unlabeled complexes (governed by Eq.12). This description is valid to good approximation because $c_{\text{H}^*}^{\text{tot}} \ll c_{\text{H}}^{\text{tot}}$ in the experiments. For clarity, we now also list these four states explicitly in the manuscript.